# XTREME-UP: A User-Centric Scarce-Data Benchmark for Under-Represented Languages

Sebastian Ruder[*]  Jonathan H. Clark[*]

Alexander Gutkin  Mihir Kale  Min Ma  Massimo Nicosia  Shruti Rijhwani
Parker Riley  Jean-Michel A. Sarr  Xinyi Wang  John Wieting
Nitish Gupta  Anna Katanova  Christo Kirov  Dana L. Dickinson
Brian Roark  Bidisha Samanta  Connie Tao
David I. Adelani[†]  Vera Axelrod  Isaac Caswell  Colin Cherry  Dan Garrette
Reeve Ingle  Melvin Johnson  Dmitry Panteleev  Partha Talukdar
Google    [†]University College London

## Abstract

Data scarcity is a crucial issue for the development of highly multilingual NLP systems. Yet for many under-represented languages (ULs)—languages for which NLP research is particularly far behind in meeting user needs—it is feasible to annotate small amounts of data. Motivated by this, we propose XTREME-UP, a benchmark defined by: its focus on the **scarce-data** scenario rather than zero-shot; its focus on **user-centric** tasks—tasks with broad adoption by speakers of high-resource languages; and its focus on **under-represented** languages where this scarce-data scenario is most realistic. XTREME-UP evaluates the capabilities of language models across 88 under-represented languages over 9 key user-centric technologies including ASR, OCR, MT, and information access tasks that are of general utility. We create new datasets for OCR, autocomplete, question answering, semantic parsing, and transliteration, and build on and refine existing datasets for other tasks. XTREME-UP provides a methodology for evaluating many modeling scenarios including text-only, multimodal (vision, audio, and text), supervised parameter tuning, and in-context learning.[1] We evaluate commonly used models on the benchmark.[2]

## 1 Introduction

The development of natural language processing (NLP) technology that serves most of world's languages is hindered by the stark lack of data for most languages (Joshi et al., 2020). While there is increasing interest in developing datasets and models for under-represented languages (ULs), existing

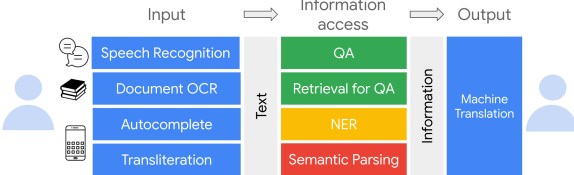

Figure 1: The tasks in XTREME-UP and their role in language technology. *Left*: enabling access to language technology; *middle*: facilitating information access as part of larger systems (question answering, information extraction, virtual assistants); *right*: making information accessible in the speaker's language.

datasets are often informed by established research directions in the NLP community (de Marneffe et al., 2021). While linguistic tasks such as syntactic parsing have become less practically relevant (Glavaš and Vulić, 2021), other impactful capabilities such as question answering or virtual assistants (Asai et al., 2021), often depend on ancillary technologies such as language ID, data filtering, automatic speech recognition (ASR), or optical character recognition (OCR) that are typically underperforming or unavailable for ULs (Caswell et al., 2020; Bapna et al., 2022; Kreutzer et al., 2022; Rijhwani et al., 2021; Khare et al., 2021). As a result, speakers of ULs are unable to reap the benefits, even if the development of models is successful.

In order to make progress on NLP for ULs, we should thus focus on evaluating models on tasks that are most likely to benefit speakers of those languages.[3] To this end, we propose XTREME-UP (Under-Represented and User-Centric with Paucal[4] Data), a benchmark focusing on evaluation of multilingual models on user-centric tasks in a scarce-

---

[*]Equal contribution. We list detailed contributions in §A.

[1]Our results in §4 indicate that few-shot in-context learning is less effective than fine-tuning on 100s of examples for ULs. We advocate for comparing such approaches directly as the community explores XTREME-UP.

[2]https://github.com/google-research/xtreme-up

[3]Speakers of ULs have needs ranging from standard NLP technology to language documentation and revitalization (Bird, 2022). Our focus is on standardized, institutional, and contact languages including dialects and non-standard language varieties spoken by large speaker populations.

[4]We borrow the term *paucal*—meaning few—from linguistics, to emphasize the scarce-data nature of XTREME-UP.

data setting.

We focus on tasks that technology users encounter in their daily lives: i) information access tasks reflecting generally useful NLP capabilities; and ii) input/output tasks that enable other technologies. We show the corresponding tasks and their role in interactions with language technology in Figure 1. Moving away from the cross-lingual zero-shot setting (Hu et al., 2020; Ruder et al., 2021), we standardize multilingual in-language fine-tuning based on the amount of data that can realistically be annotated within 8h for a language. Our results highlight the limitations of current models on ULs, demonstrate the potential of language models (LMs) to improve user-centric applications, and show the benefit of byte-based approaches.

In this work, we contribute the first massively-multilingual few-example benchmark including: **a)** newly created data for QA, OCR, autocomplete, semantic parsing, and sentence-level transliteration; **b)** new task setups for named entity recognition (NER) enabling evaluation on natural—rather than tokenized—text; and for QA and retrieval providing a more interesting setting than the gold passage (GoldP) setup while offering a lower barrier-to-entry than the full TyDi QA (Clark et al., 2020) or XOR (Asai et al., 2021) tasks; **c)** carefully-designed experimental setups, standardizing in-language fine-tuning and in-context learning and focusing on the information access scenario for ULs for ASR and MT; **d)** baseline results with commonly used subword and byte-based models.

## 2 Related Work

**Multilingual benchmarks** Some studies employ highly multilingual individual datasets for the evaluation of multilingual models, including Universal Dependencies (de Marneffe et al., 2021) or XL-Sum (Hasan et al., 2021). At the same time, there is increasing work on datasets in ULs for a variety of applications (Niyongabo et al., 2020; Winata et al., 2023; Muhammad et al., 2023). Due to their rapidly growing capabilities, NLP models are increasingly evaluated on a suite of datasets. Existing multi-task multilingual benchmarks such as XTREME (Hu et al., 2020), XGLUE (Liang et al., 2020), and XTREME-R (Ruder et al., 2021) cover 20–50 mainly high-resource languages and prioritize tasks with available data, regardless of their utility to speakers. Recently, MEGA (Kabir

et al., 2023) and BUFFET (Asai et al., 2023) evaluate in-context learning on existing multilingual tasks. In contrast, XTREME-UP focuses on under-represented languages, user-centric tasks, a more realistic scarce-data setting, and introduces new tasks and datasets.

**Multilingual evaluation** The choice of the experimental setting and aggregation metric are important considerations in multilingual evaluation. Prior work focused on zero-shot cross-lingual transfer (Hu et al., 2020), which—despite being compelling from a scientific perspective (Artetxe et al., 2020)—is less practically useful. While in-language fine-tuning has been explored before (Lauscher et al., 2020; Hedderich et al., 2020), XTREME-UP is the first to standardize the setting across tasks based on realistic annotation costs. Different frameworks aggregate performance in different ways across languages. Blasi et al. (2022) assess the utility of a task by weighting model performance based on the size of the speaker population while Khanuja et al. (2023) introduce the Gini coefficient to quantify performance disparity across languages. XTREME-UP opts for a simple average over ULs, emphasizing intuitiveness and accessibility of the results.

## 3 XTREME-UP

### 3.1 Design Principles

XTREME-UP is motivated by the following design principles:

**Under-represented languages** Following Joshi et al. (2020) we select languages in categories 1–3 (e.g., Amharic, Estonian, Kinyarwanda) as under-represented, leaving categories 4–5 as high-resource languages (e.g., English, German, Hindi). We focus on tasks with existing data in ULs and tasks where we can efficiently collect data at scale (see Appendix B for an overview of ULs in XTREME-UP).

**User-centric tasks** We focus on widely adopted user-facing tasks benefiting speakers of high-resource languages. We further break these down into two major groups: **1)** input/output tasks; and **2)** information access tasks (see Figure 1).

**Scarce data** We focus on a realistic scenario where a small amount of data is available in each UL. Mirroring reality, we do *not* restrict the amount

| | Task | Train sum over HL+ULs | Train avg. per UL | Validation sum across ULs | Test sum across ULs | # of ULs (# of HLs) | Metric | Annotation Cost minutes/example |
|---|---|---|---|---|---|---|---|---|
| Input & Output | Speech Recognition | 274,514 | 2,647 | 26,556 | 60,118 | 77(23) | CER | 0.2* |
| | Document OCR | 60 | 9★ | 447 | 452 | 7(0) | CER | 44.5 |
| | Autocomplete | 44,554 | 1,850 | 13,080 | 14,747 | 20(2) | Acc@3 | 0.3† |
| | Transliteration | 7,360 | 120 | 28,000 | 28,000 | 12(1) | CER | 2.7‡ |
| | Machine Translation | 19,877 | 120 | 34,860 | 70,000 | 70(23) | ChrF | 4.0 |
| Information Access | QA in-lang. | 59,559 | 426 | 3,656 | 3,688 | 6(3) | Span F1 | 3.0 |
| | QA cross-lang. | 22,544 | 361 | 8,199 | 12,720 | 21(6) | Span F1 | 3.0 |
| | Retrieval for QA in-lang. | 29,683 | 320 | 1,830 | 1,846 | 6(3) | MRR | 3.0 |
| | Retrieval for QA cross-lang. | 13,270 | 265 | 6,183 | 10,704 | 21(6) | MRR | 3.0 |
| | NER | 28,023 | 1,401 | 14,250 | 18,192 | 20(0) | F1 | 0.3 |
| | Semantic Parsing | 6,373 | 273 | 2,533 | 39,253 | 9(11) | EM | 2.0 |

Table 1: The tasks in XTREME-UP. For each task, we show both the sum of training examples across all languages—to give some insight into training scale—and the average number of training examples *for each under-represented language*—to highlight the challenge of the scarce-data learning scenario. XTREME-UP does not limit supervised training data in high-resource languages (HLs) while each under-represented language (UL) has a maximum of 8 hours of annotation effort in its training split; see last column for estimated annotation effort. We also show the sum of validation and test examples across ULs as XTREME-UP evaluates only on ULs.

*Average time for read speech. †Based on mean typing speed (Dhakal et al., 2018) and average sentence length in Universal Dependencies (de Marneffe et al., 2021). ‡An annotated example can be used for both task directions. ★For document OCR, each example is a whole page; for example, 9 pages of training images per language corresponds to approximately 290 lines of output text per language.

of training data available in high-resource languages, but rather provide only as many labeled training examples as can be annotated in a realistic amount of time for ULs (see Section 3.2).

**Efficiency** We focus on massively multilingual evaluation settings that can still be run efficiently with a modest amount of compute.

**Text-centric, yet multi-modal** We focus on tasks that can be tackled using textual data alone and provide baseline systems that do so. We frame multi-modal tasks (OCR and ASR) so that natively multi-modal models can be evaluated fairly alongside text-only models. We accomplish this by releasing original audio, image, and text model inputs while also providing baseline system output that can be fed to second-stage text-only systems. We hope to see fully multi-modal models take up this challenge over the coming years.

We provide an overview of the tasks in XTREME-UP in Table 1. We discuss motivation and high-level information in the next section and provide more details for each task in Appendix D.

### 3.2 How much data?

To ensure a realistic amount of training data, we limit the training data in each task per language to the number of examples that can be annotated in 8 hours. We believe this reflects the real difficulty of annotating training and evaluation data for a very large number of languages. In this way, we design for the *task first*. For each task, we estimate how

long it takes to annotate a single example for a trained annotator.[5] We base our estimates on prior work and our own annotation efforts.[6] We show the data annotation time estimates in Table 1. For tasks with larger training datasets, we sub-sample the available data accordingly. Table 1 shows the sub-sampled data sizes. We show the input and output format of each task in Table 2. We provide an example instance of each task in Appendix C.

### 3.3 Input / Output Tasks

**Automatic speech recognition (ASR; D.1)** The goal of ASR is to transcribe speech into human-readable text. It thus serves as a fundamental step for enabling natural language understanding applications on speech input. In many contexts, users may strongly prefer to speak rather than type and so high-quality ASR is an enabling factor for such interactions. We employ the FLEURS dataset (Conneau et al., 2023) consisting of recordings in 102 languages for sentences from FLORES-101 (Goyal et al., 2022), which were translated from English Wikipedia to 101 languages. We evaluate on 77 under-represented languages.

**Optical character recognition (OCR; D.2)** OCR, the process of converting text from images

---

[5] For simplicity, we estimate the annotation time for labeling only, ignoring factors such as training annotators, data processing, data validation, interface design, etc. We note that unlabeled data may not be available for certain ULs (Nekoto et al., 2020) and its creation may require tools such as keyboards, which may not be available in all languages.

[6] For autocomplete, we calculate average writing time.

| Task | Input | Output |
|---|---|---|
| Speech Recognition | Speech audio | Transcription |
| Document OCR | Page image | Transcription |
| Autocomplete | Sentence prefix | Completed word |
| Transliteration | Source script sentence | Target script transliteration |
| Machine Translation | Source language sentence | Target language translation |
| In-language Retrieval for QA | Source language question | Source language answer passage |
| In-language QA | Source language question, title, and passage | Source language answer span (or "No answer") |
| Cross-language Retrieval for QA | Source language question | English answer passage |
| Cross-language QA | Source language question, English title and passage | English answer span (or "No answer") |
| NER | Sentence | Entity mentions with labels |
| Semantic Parsing | Assistant query | Structured parse with intent, slots |

Table 2: The input and output format of each task in XTREME-UP. Tasks are generally text-in, text-out with a few exceptions. See Appendix C for task examples.

into machine-readable formats, is used in a wide range of applications, from extracting data only available in paper books (Rijhwani et al., 2020) and imaging legal documents (Singh et al., 2012), to improving accessibility for people with low vision (Mowar et al., 2022). It is especially important for under-represented languages, where both training data and content that users may wish to access may not be available as digital text on the web.

We create a dataset that aims to fill the gaps in previous work in OCR for ULs (see Appendix D.2) by focusing on larger-scale, typologically diverse, and user-centric data. Our dataset contains transcriptions for books in seven languages: Amharic (am), Bengali (bn), Kannada (kn), Myanmar (Burmese; my), Sanskrit (sa), Sinhala (si), and Swahili (sw). The books domain is the primary use-case for a large number of downstream users, but is one of the most challenging for OCR models (Rigaud et al., 2019). The dataset consists of transcriptions of entire pages and thus enables leveraging the full context understanding capabilities of large language models.

**Autocomplete (D.3)** Autocomplete (or predictive text), i.e., predicting the rest of a word a user is typing, is a useful technology that speeds up human-computer interaction (Anson et al., 2006). As such, autocomplete has become a technology

that users have come to expect and rely on for input in high-resource languages. The standard next word prediction task (Sundermeyer et al., 2012) does not accurately reflect this practical setting as it relies on predicting entire units (words, subwords, or characters); similarly, perplexity-based evaluation makes comparisons across segmentations and languages difficult (Mielke, 2019) and ignores threshold effects associated with top-k predictions in a user interface (Tam and Wells, 2009).

To fill this gap, we introduce a new autocomplete task that unifies character, subword, and token-level LM settings by focusing on a "word" as the predictive unit. Models are required to complete the next word based on a left context of $N$ words and an optional character n-gram prefix. We use accuracy@3 for evaluation to reflect the requirement of displaying a limited number of candidates to the user. We process high-quality natural language data from Universal Dependencies (de Marneffe et al., 2021), which we deduplicate against mC4 (Xue et al., 2021), the most common multilingual pre-training corpus in order to test models' predictive rather than memorization capabilities.

**Transliteration (D.4)** Transliteration is the conversion of text between writing systems (Wellisch, 1978). Unlike translation, it does not change content but only script. Transliteration is important because it allows users to type in their preferred script (e.g., Latin script) even if it is different than their preferred display script (e.g. Devanagari) and is used internally by many machine translation systems to rewrite names from different scripts.

We extend the Dakshina dataset (Roark et al., 2020), which provides romanizations of Wikipedia sentences written in the native scripts of 12 South Asian languages, with: a) romanizations of native script Wikipedia for one new language (Amharic); and b) transliteration to a third script (Shahmukhi) for one already covered language (Punjabi). The resulting task covers 13 languages for which transliteration occurs from the Latin script to the native script, and vice versa, and between Shahmukhi, Gurmukhi, and Latin for Punjabi.

**Machine translation (MT; App. D.5)** MT is an important technology for users of ULs wishing to read text written in a different language. However, most current approaches require large amounts of parallel training data to achieve good performance, which are often not available for ULs (Had-

dow et al., 2022). We focus on the information dissemination scenario where content from high-resource languages (including from tasks such as cross-lingual QA) is translated to enable information access by common users; as such, XTREME-UP includes translations from English into 93 languages, covering a wide range of high-resource and UL languages. Only 39 ULs are used for evaluation; the high-resource languages are included to allow for transfer learning.[7] The dataset is adapted from FLORES-101 (Goyal et al., 2022), repurposing half of the dataset's original development set as a training set. See §6 for a detailed discussion of how we distinguish freely-available unsupervised data versus purpose-annotated supervised data in XTREME-UP.

### 3.4 Information Access Tasks

**Question Answering (D.6)**   Question answering enables responding to natural language questions with answers found in text. We focus on the *information-seeking* scenario (Kwiatkowski et al., 2019) where questions are asked without knowing the answer. Information-seeking question-answer pairs tend to exhibit less lexical and morphosyntactic overlap between the question and answer since they are written separately.

We include two variants of the task: In **in-language QA**, both question and passage are in the same language. We obtain original questions and passages from TyDi QA (Clark et al., 2020). For **cross-language QA**, the question is in the user's native language while passage and answer are in a language with a large amount of answer content available (English). We use examples from TyDi XOR (Asai et al., 2021) in 7 languages. We additionally collect new data in 23 new Indic languages for cross-lingual QA by professionally translating questions and answers from existing Indic languages in XOR QA. This methodology mitigates the issue of translating Western-centric English data to locales with different topical interests. Cross-lingual QA is especially important for ULs since they lack plentiful in-language answer content on the web.

In XTREME-UP's QA task, a system is given a question, title, and a passage and must provide the answer—if any—or otherwise return that the question has "no answer" in the passage.[8] To this end, we generalize the gold passage (Clark et al., 2020) setting, augmenting it with negative examples. These negatives are obtained from (a) passages within the same article as a passage containing the answer and (b) question-answer pairs from the full TyDi QA dataset where no answer was found in the candidate Wikipedia article. The data is split into training, validation, and test splits in such a way to avoid deduplication and overlap of splits, even across our various QA tasks. [9]

**Retrieval for QA (D.6)**   Within the information-seeking QA scenario, the above core QA task assumes answer candidate passages as an input. In practice, a passage retrieval system for question-answering allows for the extraction of relevant text from a vast text corpus. The retrieved passages can then be used by a question-answering system to extract or generate an answer to the user's question. In XTREME-UP, we separate retrieval into two distinct tasks, **in-language retrieval** and **cross-language retrieval**. For in-language retrieval, both the questions and passages are in the same language. The preparation of negatives, deduplication, and splits are identical to the QA task above. For validation and test, we create an index of 271k in-language passages (447k English passages for the cross-language task) making for a small enough index for efficient experimentation, while containing distractors that make for a challenging task, since these distractors are drawn from the same articles containing the target passages.

**Named entity recognition (NER; D.7)**   NER is an important capability for information access systems that users depend on with applications ranging from recognizing requests for entity lookups to performing information extraction to populate the knowledge graphs that handle those requests. NER is also a capability needed in spell-checking and localization systems (Li et al., 2020).[10] Identifying entities in ULs poses challenges due to the use of different scripts, lack of capitalization, different numerical representations, etc. We build on MasakhaNER (Adelani et al., 2021) and MasakhaNER 2.0 (Adelani et al., 2022), two large

---

[7]Our baseline results were trained only on the 39 UL pairs for efficiency.

[8]This follows SQuAD v2 (Rajpurkar et al., 2018).

[9]This turns out to be non-trivial given the different splits strategies across the various datasets and our decision to create a train, validation, and test set even where only a train and validation set were previously available for public download.

[10]We emphasize the word *capability* here since we recognize that stand-alone NER *systems* may not be strictly necessary in the long run; however, the capability of recognizing and properly handling entities will remain.

NER datasets in African languages, which provide data in the standard CoNLL tokenized format (Tjong Kim Sang and De Meulder, 2003). In order to enable evaluation in a setting that is closer to the real world, we automatically map the annotated spans to the original raw text. The combined data with byte-level span annotations—termed MasakhaNER-X—covers 20 languages.[11]

**Semantic parsing (D.8)**    Semantic parsing is the task of mapping a natural language utterance to a logical form or a structured interpretation that can be executed by a system such as a virtual assistant. This task is especially timely as users will increasingly want to turn their interactions with assistants and chat-like dialog systems into actions on external systems, which require API calls; this capability is what the semantic parsing task evaluates.

We adapt the test split of MTOP[12] (Li et al., 2021) with professional translators/annotators to 15 languages: Amharic, Belarusian, Bengali, Brazilian Portuguese, Finnish, German, Hausa, Hungarian, Japanese, Russian, Swahili, Tamil, Turkish, Yoruba, and Zulu. Together with the original MTOP languages, the new MTOP++ dataset covers a total of 20 languages. Differently from MTOP, we collect localized data (i.e., Western-centric entities are replaced with more culturally relevant entities for the target language), following recent trends in multilingual benchmarking (Lin et al., 2021; Ding et al., 2022; Majewska et al., 2023).

We also extend MTOP to three widely spoken but under-represented Indic languages in a code-switching setting: Hindi-English, Bengali-English and Tamil-English. We automatically convert the test-split of MTOP to code-mixed utterances using PaLM (Chowdhery et al., 2022) and run human verification on such utterances.

### 3.5   Overall Evaluation

For each task, we evaluate model performance by computing a task-specific score. We employ character-level metrics such as character error rate (CER) and character n-gram F-score (chrF; Popović, 2015) rather than their word-level counterparts as they enable more fine-grained evaluation and are better suited to morphologically rich languages. We obtain a final score by averaging the

| Model | Eval setting | # of params | Vocab units | % of non-en pre-train data |
|---|---|---|---|---|
| mT5-Base | FT | 580M | Subwords | 94.3 |
| ByT5-Base | FT | 580M | Bytes | 94.3 |
| Flan-PaLM | ICL | 62B | Subwords | 22.0 |

Table 3: Additional information on baseline models including the setting in which we evaluate them (fine-tuning vs in-context learning), their size, their vocabulary, and the fraction of non-English pre-training data.

scores of all tasks. For each task, we only average performance over ULs (discussed in §3.1). For metrics such as character error rate (CER) where lower is better, we invert the scores before averaging scores across tasks. For mean reciprocal rank (MRR), which is in the 0.0–1.0 range, we renormalize it to the 0–100 range before averaging. While this scalar provides a quick overall impression of a system's quality across a broad range of tasks, it is not a substitute for analyzing performance on individual tasks, languages, or types of examples.

## 4   Experiments

### 4.1   Experimental setting

**Multilingual fine-tuning**    In contrast to prior benchmarks that focus on zero-shot cross-lingual transfer from English, XTREME-UP focuses on the more realistic scenario of fine-tuning on a small amount of data in the target language. To make this scenario scalable in a massively multilingual setting, XTREME-UP fine-tunes a single model on the combined training data across the available languages for each task. The data for each language is sub-sampled to emulate data sizes that can be realistically annotated within a reasonable time frame (see §3.2).

**In-language in-context learning**    We also provide a 5-shot in-context learning setting where a model is provided with an English instruction and 5 exemplars in the target language in order to evaluate the progress on few-shot learning with large models for ULs. We provide the instruction for each task in Appendix E.[13]

---

[11]We remove the Fon and Hausa subsets of MasakhaNER 2.0 due to quality issues in the annotated data.

[12]Other datasets (see App. D.8) were not yet available at the start of the project and do not focus on ULs.

## 4.2 Baselines

We provide results on a handful of baseline systems that have already been developed by the research community. Given that our focus in this paper is on the dataset and task setup rather than system building, we do not focus on offering novel modeling types nor do we exhaustively evaluate all possible models; rather we view these results as estimating a starting point from some well-known modeling approaches and seeding contributions from the broader research community.[14]

**Multilingual fine-tuning baselines** For the main experimental setting of multilingual fine-tuning, we provide the following baselines: **mT5-base** (Xue et al., 2021) and a subword-based multilingual encoder-decoder model; **ByT5-base** (Xue et al., 2022), a byte-based multilingual encoder-decoder model.

**In-context learning baseline** For the in-context learning setting, we employ **Flan-PaLM** (Chung et al., 2022), an instruction-tuned version of PaLM (Chowdhery et al., 2022). We provide additional information on the baseline systems in Table 3.

To offer baseline systems that allow experimentation with text-only models, we use upstream models to provide initial output for ASR and OCR, and present text-based baselines that use these as inputs. We expect these baselines to give way to fully multi-modal models as research progresses. These initial ASR and OCR outputs should be seen as part of a baseline system, *not* part of the XTREME-UP benchmark itself. For ASR, we augment the data with predictions of the state-of-the-art Maestro-U (Chen et al., 2023) and then use a downstream text model to improve the outputs (Bassil and Alwani, 2012). Similarly, for OCR, we use the off-the-shelf Google Vision OCR[15] to get first-pass outputs, and train language models to improve them (Dong and Smith, 2018; Rijhwani et al., 2020).

**Infrastructure** Models were trained using `seqio` and `t5x` (Roberts et al., 2022) on TPUs (Kumar et al., 2019; Pope et al., 2022).

## 4.3 Results

We show the baseline results in Table 4.

**Byte-based models outperform subword-based on ULs.** The byte-based ByT5 outperforms the subword-based mT5 across most tasks. Gains are particularly pronounced for tasks that require dealing with information on the character level such as autocomplete and transliteration and for predicting information on the word level such as for NER and semantic parsing. These results demonstrate that as we train and evaluate our models on under-represented languages, standard modeling choices such as subword representations fall short.

**In-context learning underperforms fine-tuning on limited data.** The Flan-PaLM model generally performs worse than fine-tuned models, despite being much larger. Nevertheless, it achieves reasonable performance on machine translation, which is likely reflected in the pre-training data. On other tasks, however, it fails to reliably apply its English-centric knowledge to ULs. Despite fine-tuned models performing relatively well on NER, the in-context learning model is unable to consistently generalize to the task in a few-shot setting in under-represented languages. On semantic parsing, the model fails to generalize to the large number of domain-specific intents and slots using standard prompting in ULs.[16] The autocomplete tasks in particular demonstrate the lack of robust cross-lingual information in the English-centric PaLM model: it struggles to complete a sentence given a character prefix and fails to reliably convert between different scripts in the same language. XTREME-UP thus provides a strong challenge to test the generalization abilities of in-context learning methods to ULs.

**There is a lot of headroom left to improve performance on ULs.** Overall, across all tasks there is still a considerable amount of headroom left. For ASR, OCR and transliteration, around 10% of characters are still incorrectly predicted. On autocomplete, models only make the correct prediction in about one fourth of all cases. For MT, on average

---

[13]The choice of prompt and exemplars can have a significant impact on performance (Zhao et al., 2021a,b). We provide a single instruction and set of exemplars per task and language for replicability and leave the search for better instructions and exemplars to future work.

[14]XTREME-UP offers a **public results tracker** for use in tracking the community's progress on XTREME-UP. We conceptualize these results not as a competition, but as offering insights about different models and their trade-offs, each justifying and explaining how it should be compared to the others and how it informs the research landscape. Submissions can be made via self-service git pull requests.

[15]https://cloud.google.com/vision/docs/ocr

[16]We leave the exploration of multilingual adaptive prompting and dynamic exemplar selection (Drozdov et al., 2023) methods to future work.

|  | Input & Output Tasks | | | | | Information Access Tasks | | | Semantic Parsing |
|  | ASR | OCR | Autocomplete | Transliteration | MT | QA | Retrieval | NER | |
|  | CER↓ | CER↓ | Acc@3↑ | CER↓ | chrF↑ | F1↑ | MRR↑ | F1↑ | EM↑ |
| *Multilingual fine-tuning* | | | | | | | | | |
| mT5-Base | 8.5 | (11.1)★ | 12.7 | 37.6 | 22.5 | 59.7 (74.9 / 44.6) | 0.23 (0.41 / 0.07) | 74.0 | 21.8 |
| ByT5-Base | 8.2 | (11.1)★ | 27.6 | 14.6 | 26.9 | 71.4 (82.3 / 60.5) | 0.29 (0.45 / 0.18) | 84.0 | 37.5 |
| *In-context learning (5-shot)* | | | | | | | | | |
| Flan-PaLM-62B | 23.2 | — | 0.0 † | 77.4 | 32.1 | 22.9 (20.9 / 24.9) | — | 12.9 | 0.1 |

Table 4: Overall results of baselines across all XTREME-UP v1.0 tasks for the test split. Scores on XTREME-UP average over evaluation scores of *under-represented* languages. QA and retrieval performance is the average of in-language and cross-language settings (indicated in brackets as in-language / cross-language). For OCR, we do not apply any additional models (mT5 nor ByT5) on top of the baseline OCR system; we show these results in parentheses. We do not attempt in-context learning (ICL) results for retrieval since ICL is typically only used for text-in, text-out use cases. ★For OCR, we use the Google OCR API. †For autocomplete, while we observe reasonable performance on English completions, we find the model typically does a very poor job outside English.

only about a third of n-grams in the hypothesis are also present in the reference, and vice versa. For QA and retrieval, there are large performance differences between in-language and cross-language settings and much headroom still left. On NER, models perform relatively well but are still far from perfect performance on the task. Finally, on semantic parsing models are only able to produce the correct output in around a third of all cases.

## 5 Analyses

**Lowest-performing languages** Models generally perform poorly on African languages. On transliteration, models perform relatively worst on the newly added Amharic language. On NER, which covers only African languages, performance is lowest for Amharic—likely due to its different script—and the extremely under-represented Ghomálá'. Similarly, translation models underperform in Amharic and Yoruba. On ASR, the lowest-performing languages are Yoruba but models also struggle with other languages such as Gaelic, and many South Asian languages such as Lao, Khmer, and Burmese.

**Task-specific observations** ByT5 provides the best performance while the size of the model does not seem to impact performance much. Several aspects of the data lead to higher error rates in transliteration: the model struggles with input in the Perso-Arabic script and to produce output in Latin based on a different script. For autocomplete (see Appendix D.3), our analyses indicate that models perform better on text that uses the Latin script.

## 6 Recommendations

**Use of splits** XTREME-UP offers a train, validation, and test split for each task. We recommend using the training split for learning the parameters of your model or as exemplars for in-context learning while iteratively checking your progress on the validation (i.e. development) split. The test split should *not* be used for iterative evaluation of your models or other sorts of hill-climbing; instead, it should be reserved for reporting your results and comparing *after* you have finished development on your models. Experiments that follow this customary scientific rigor should expect to show better generalization and less overfitting to the test split.

**Use of additional pre-training data** One potential confounder for results along different pre-trained models is the variation in pre-training data; where this data overlaps with the targets (outputs) in XTREME-UP validation and test splits, results can be artificially inflated, providing a sense that results are better than they are in reality—if the validation or test data leaked into the pre-training data via contamination during large-scale data scraping, then it's unlikely that the system would truly perform as well for new unseen inputs. Therefore, we recommend that when researchers modify the pre-training data for a model, they explicitly report overlap (contamination) between the targets of the XTREME-UP validation/test splits and their pre-training corpus.[17]

_______________

[17]We recognize that this is a very large-scale undertaking, requiring a fairly large amount of compute. As such, we suggest that it's may only be needed when making claims that compare systems (e.g. that the system with possibly-

**Use of additional supervised data** It is entirely possible that the community will find creative ways to improve models based on supervised data not included with XTREME-UP. However, researchers should bear in mind how this might affect the comparability of their results with other models. The following axes should be considered:

1. *Any* additional data from high resource languages is always allowed in the XTREME-UP setting.
2. Supervised data (e.g. parallel data for MT) harvested from the web, religious, books, and other opportunistic sources will typically be out-of-domain and is therefore admissible; conversely, supervised data from ULs from highly similar tasks or domains should generally be considered against the spirit of the XTREME-UP benchmark.
3. Monolingual data from UL is admissible with the caveat that one should measure overlap with targets, as discussed above.

**Avoid off-the-shelf MT systems** Data augmentation via automatically translating high-resource supervised data to languages with less supervised data has proven a very effective means of improving system quality. However, it is not necessarily realistic to use a pre-existing MT system (e.g. an API or an open-source model) since those systems have typically been trained on a large amount of parallel data—or at least unknown data. This means that additional supervised data would then be leaking into the experimental setup, which is otherwise intended to reflect the reality that most under-represented languages have very little supervised data. If data augmentation via translation is used, we encourage researchers to report the parallel data sources used and argue why this experimental setup is realistic—or to clearly point out such usage in their experiments as an unavoidable confound and discuss the limitations this sets on what conclusions can be drawn about how results will extrapolate to the breadth of under-represented languages.

In all cases, researchers should rigorously report what additional data was used and how; each use case comes with its own considerations and,

above all, researches should make a well-reasoned argument that their use of data (i) does not artificially inflate evaluation scores and (ii) reflects a real-world scenario of finding and applying data.

## 7 Conclusion

We have presented XTREME-UP, a multilingual benchmark distinguished by its being (i) scarce-data, (ii) user-centric, and (iii) focused on under-represented languages. The benchmark contains input modalities of text, images, and audio while still allowing experimentation with text-only models. We hope this benchmark will be useful in accelerating research that is useful to speakers of under-represented languages and in highlighting both the progress and limitations of current models of language.

## Limitations

The dataset presented in this work does not represent all of the world's languages nor all of the under-represented languages. While we have made efforts to include languages and dialects across a broad variety of geographic regions and language families, it was not feasible to locate or create data in the same set of languages across all tasks. Since this is a data-focused paper, we present modeling results on a few strong modern models; this is not an exhaustive exploration of how all current models may perform on this dataset. We look forward to exploring more under-represented languages as more data becomes available.

## Acknowledgements

We thank Slav Petrov, Jason Riesa, Raphael Hoffmann, Dipanjan Das, Clara Rivera, Chris Alberti, Machel Reid, and Timothy Dozat for helpful discussions and feedback. We are grateful to Noah Constant for a review of a draft of the paper. We also gratefully acknowledge the contributions of the researchers who built the datasets that have gone into XTREME-UP; we recommend that all component datasets be cited individually when using XTREME-UP in a paper such that dataset authors (many of whom are not authors of this article) receive credit for their work and so that those original sources remain easily discoverable in the literature.

---

contaminated pre-training data is equivalent, better, or almost as good as some other system). Note, this analysis only needs to be done once for each pre-training corpus (e.g., once for mC4) and it is very likely that organizations with enough compute to pre-train a new model on a new corpus would also have sufficient compute to calculate overlap.

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

# A  Contributions

In this section, we provide more detail about the contributions of each author.

## General overview

**Project leads**  Sebastian Ruder, Jonathan Clark

**Primary contributors and task owners**  Alexander Gutkin, Mihir Kale, Min Ma, Massimo Nicosia, Shruti Rijhwani, Parker Riley, Jean-Michel Sarr, Xinyi Wang, John Wieting

**Major contributors**  Nitish Gupta, Anna Katanova, Christo Kirov, Dana Dickinson, Brian Roark, Bidisha Samanta, Connie Tao

**Supporting contributors**  David Adelani, Vera Axelrod, Isaac Caswell, Colin Cherry, Dan Garrette, Reeve Ingle, Melvin Johnson, Dmitry Panteleev, Partha Talukdar

## By Task

**ASR**  Min Ma

**Autocomplete**  Jean-Michel Sarr, Vera Axelrod, Colin Cherry, Sebastian Ruder, Jonathan Clark

**MT**  Parker Riley, Isaac Caswell, Colin Cherry, Jonathan Clark

**NER**  Sebastian Ruder, David Adelani, Dan Garrette

**OCR**  Shruti Rijhwani, Dana Dickinson, Reeve Ingle, Dmitry Panteleev, Sebastian Ruder

**QA**  Mihir Kale, John Wieting, Nitish Gupta, Partha Talukdar, Jonathan Clark

**Retrieval**  John Wieting

**Semantic parsing**  Massimo Nicosia, Bidisha Samanta, Partha Talukdar

**Transliteration**  Alexander Gutkin, Anna Katanova, Christo Kirov, Brian Roark

**Evaluation framework and public results tracker**  Xinyi Wang

**Program management**  Connie Tao

**Fine tuning and modeling**  Alexander Gutkin, Mihir Kale, Min Ma, Massimo Nicosia, Shruti Rijhwani, Parker Riley, Jean-Michel Sarr, John Wieting, Sebastian Ruder, Jonathan Clark

**Data processing**  Alexander Gutkin, Mihir Kale, Min Ma, Massimo Nicosia, Shruti Rijhwani, Parker Riley, Jean-Michel Sarr, Xinyi Wang, John Wieting, David Adelani, Vera Axelrod, Isaac Caswell, Colin Cherry, Dan Garrette, Reeve Ingle, Dmitry Panteleev, Sebastian Ruder, Jonathan Clark

**Data collection**  Massimo Nicosia, Bidisha Samanta, Nitish Gupta, Anna Katanova, Christo Kirov, Dana Dickinson, Brian Roark

**In-context learning**  Sebastian Ruder

**Benchmark design**  Jonathan Clark, Sebastian Ruder, Melvin Johnson

## B   Language Coverage

We provide an overview of the under-represented languages in XTREME-UP in Table 5. For each language, we indicate a) the ISO 639-1 code (or ISO 639-3 code if the former is unavailable); b) its language family according to Glottolog (Nordhoff and Hammarström, 2011); c) the number of datasets in XTREME-UP including the language; d) its resource level based on the taxonomy of Joshi et al. (2020) (0 is least and 5 is highest-resourced); and e) which tasks include the language.

## C   Task Examples

We provide an example instance of each task in Table 6.

## D   Data cards

### D.1   ASR

#### D.1.1   Task description

Automatic speech recognition (ASR) transcribes speech inputs into human-readable text, serving as a fundamental step for various speech language understanding applications. The transcripts are often calibrated with some pre-trained language models to produce the final outputs. In this paper, we build the ASR benchmark in this way: first, transcribe input audio into text with a pre-trained speech recognition model; then calibrate the transcripts by fine-tuning pre-trained language models on paired transcripts and ground truths.

#### D.1.2   Data creation

Experimented on the FLEURS corpus (Conneau et al., 2023), we use Maestro-U (Chen et al., 2023) to generate the ASR transcripts. For the pre-trained language models, we choose mT5-base (Xue et al., 2021) and ByT5-base (Xue et al., 2022) models. We paired the ASR transcripts with the ground truths to fine-tune the mT5 or ByT5 models. The average character error rate (CER) of Maestro-U is 8.28% across 102 languages, providing a strong baseline. Therefore, we build the ASR benchmark in a selective way: first, we compare the Maestro-U baseline CER on the dev set with the CER obtained by fine-tuned mT5 or fine-tuned ByT5. If the fine-tuned result is better, we choose the fine-tuned model for the language to rescore its test set; otherwise, we keep the baseline Maestro-U results for the test.

#### D.1.3   Data structure

We followed the data split of train, dev, and test sets in FLEURS, and filtered out the examples where Maestro-U prediction is empty (i.e., all the deletion errors). The pairs of transcript and ground truth are saved in `jsonl` and `tsv` format.

The individual language datasets are mostly distinguished by the language and region BCP-47 codes, e.g., the `kam_ke` code represents Kamba language spoken in Kenya. In some cases, when multiple writing systems are available for a language, the ISO 15924 script code is used as well, as is the case with the code `sd_arab_in` that denotes Sindhi as spoken in India and recorded using Arabic script, as opposed to its Pakistani counterpart.[18]

#### D.1.4   Data statistics

The FLEURS dataset contains about 1.4k hours of audio in total for 102 languages. The training data contains 271,488 examples across 102 languages, average length per utterance is about 20 tokens. There are 34,661 examples in the validation (dev) set, and 77,943 examples in the test set.

#### D.1.5   Experiments and Discussion

We compared fine-tuned mT5-base and ByT5-base baselines, which were built on TPU. In addition, we explored the compute efficient fine-tuning on GPU, using a mT5-small model as pre-trained model. The three models took 4500, 6500 and 4000 steps to converge, respectively. We report the character error rate for the predicted transcripts by the fine-tuned models against the one for the Maestro-U baseline, which is 8.28% on average for 102 languages – a quite strict baseline. We observed small gains through fine-tuning with different pre-trained models, as shown in Table 7.

It is observed that ByT5 yields better fine-tuned results than mT5, indicating that byte is a better modeling unit when it comes to textual data of various writing systems. By calculating the average CER for 24 high-resourced language group and 78 low-resourced language group respectively, we find that both mT5 and ByT5 fine-tuned models can reduce CER from 6.40% baseline to 6.36% for high-resourced languages, while ByT5 on its own can further improve CERs for low-resourced languages from 8.86% baseline to 8.80%.

Fine-tuned ByT5 also generalized well on languages which were not seen in the pre-training

---

[18]The `es_419` code represents Latin American Spanish.

phase. With a limited amount of fine-tuning data, ByT5 can improve baseline on the group of unseen languages, especially on Umbundu (umb_ao, -14% CER Relative). Even though only Romanized Chinese is used to pre-train ByT5, the fine-tuned ByT5 outperformed baselines for both Mandarin (in simplified Chinese, cmn_hans_cn), and Cantonese (in traditional Chinese, cmn_hant_hk).

## D.2 Optical character recognition (OCR)

### D.2.1 General information

**Dataset title**  UL-OCR

### D.2.2 Related work

While most existing datasets focus on higher-resourced languages (Nayef et al., 2017; Rigaud et al., 2019), there has been recent interest in developing OCR for ULs. This includes the creation of a dataset for OCR on endangered languages (Rijhwani et al., 2020) and a synthetic dataset for 60 languages (Ignat et al., 2022).

### D.2.3 Data creation

We retrieve books that are in the public domain on Google Books. These are historic books, where the copyright has expired, as well as more recent and public-domain books, used in this dataset with approval from their publishers. We focus on languages with diverse scripts, for which no existing OCR dataset is currently available. We observe that many public-domain books in such languages are religious or linguistic in nature and were created for missionary purposes. In order to identify a diverse set of high-quality books, we first conduct an annotation task where we ask annotators to look at pages of a book and assign whether it is a) not in the target language, b) religious, c) consisting mainly of tables/other structured formatting, d) linguistic (e.g., a dictionary or grammar book), e) not intelligible, or f) low quality. Based on this annotation, we filtered out some languages that did not have a sufficient amount of high-quality public-domain books available. After filtering, the dataset contains annotated documents in seven under-represented languages – described in detail in Section 3.3.

## D.3 Autocomplete

### D.3.1 Task description

Autocomplete (or predictive text), i.e., predicting the rest of a word a user is typing, is a useful technology that speeds up human-computer interaction. However, while language modeling (LM) is a core natural language processing (NLP) task, current LM evaluation does not address the practical constraints of human-computer interaction and current LMs are not directly useful for autocomplete in under-represented languages.

In order to evaluate multilingual models on an evaluation setting as close as possible to the real-world usage of autocomplete, we curated the Universal Dependencies (UD) dataset (Nivre et al., 2020; de Marneffe et al., 2021) according to a set of high level principles that we describe in the section below.

### D.3.2 Data creation

The original UD dataset was filtered to better fit the user centric paradigm proposed. We removed a) treebanks using only ancient data, for example liturgical text written in Latin, Ancient Greek or Sanskrit; b) languages with fewer than 100 speakers like Akuntsú; c) signed languages like the Swedish Sign Language; d) highly domain-specific content like for instance SiMoNERo (Mititelu and Mitrofan, 2020) which contains texts from three medical subdomains: cardiology, diabetes, endocrinology; e) languages that are "high resource" by XTREME-UP standards with the exception of English which we kept for prototyping; f) languages that do not have all three of: training, validation and test sets: g) languages with fewer than 1000 examples when combining training and validation set.

The resulting corpus features 23 languages: Basque, Belarusian, Bulgarian, Danish, Eastern Armenian, English, Estonian, Galician, Scottish Gaelic ,Greek, Hebrew, Icelandic, Indonesian, Irish, Latvian, Lithuanian, Nigerian Pidgin, Romanian, Slovak, Slovenian, Ukrainian, Urdu, and Uyghur.

### D.3.3 Data structure

A data instance has two fields, input and target, for instance {input: "en_-We look f$", target: "forward"}. The input field is composed of a prefix "en_-" to indicates the language to the model and a context sentence: "We look f$". The target field is the word to predict. We normalize all text with Unicode NFKC normalization (Whistler, 2021).

**Annotation process**  In the following, we describe how the example described above is generated from the source data. The original sentence is "We look forward to your active participation to make this forum an exciting meeting place for like minded individuals." The steps are: a) The context sentence including the target can have at most 10

words. A random word of more than 5 characters is chosen to be the target. b) A target context is sampled from the target and added to the context. In this example it is the character "f". The sample rule is to select a number of characters that can vary between 0 to the number of characters in the target minus three. In our example, the target "forward" could be sampled from "" to "forw". c) A specific token "$" is added just after the target context.

### D.3.4 Data statistics

We sampled up to 2,000 examples from each language's training set, 1,000 examples from validation, and 1,000 examples from test. This prevents the languages from having disproportionately more data; where the original sets were smaller than these targets, we used all available data. We display the language statistics in Table 8. Note that these experiments are done on a preliminary dataset and **not** the final release version of XTREME-UP.

### D.3.5 Experiment

We compared mT5 (Xue et al., 2021) and ByT5 (Xue et al., 2022), two state-of-the-art multilingual pre-trained LMs that are based on subwords and bytes respectively. The models were fine-tuned for 10 epochs on autocomplete training set, moreover. We used two metrics: top-3 word accuracy (Acc@3) and chrF: character $n$-gram F-score (Popović, 2015).

### D.3.6 Results

We observe that ByT5 achieve better performance than mT5 for both Acc@3 and chrF on the autocomplete task as it is displayed in Table 9. Also ByT5 require less than half the time to fine-tune on the training set (45 minutes) compared to mT5 (1 hours and 30 minutes).

### D.3.7 Analyses

Based on Acc@3 and chrf, the most challenging languages for mT5 are Eastern Armenian ((hy)) and Uyghur (ug) respectively. Whereas Nigerian Pidgin is the (pcm) and Scottish Gaelic are the easiest languages. For ByT5, whether we consider Acc@3 or chrF, the most challenging language is Uyghur, and the easiest language is Galician (gl). Yet, these extremes only offer a qualitative comparison of mT5 and ByT5. Next, we investigate four questions around model performance: a) Do mT5 and ByT5 have the same cross-lingual generalization pattern? b) Do some languages yield higher scores because autocompletion guesses the

same words? c) Do some languages yield higher scores because they have a smaller vocabulary in their corpora? d) Does similarity to the Latin alphabet impact models' performance? We test several hypotheses below, considering a relationship to be significant when the $p$-value is under $0.05$.

**Do mT5 and ByT5 have the same cross-lingual generalization pattern?** mT5 and ByT5 have the same cross-lingual generalization pattern if the difficulty to generalize to a new language is the same for both models relatively to other languages. In other words, if models' performance are ranked similarly, they share the same cross-lingual generalization pattern. To evaluate this hypothesis we computed the Spearman's rank correlation between mT5 and ByT5 Acc@3. We got a Spearman's rank correlation of 0.69 with $p$-value $< 0.001$. This means that the two models have a high degree of relative agreement, in other words, if a new language is added, there is a high chance that the language is going to be challenging or not for both mT5 and ByT5.

**Do some languages yield higher scores because autocompletion guesses the same words?** If our dataset in given language over-represents a word to predict, then the model might have misleadingly good performance by always predicting the same word. This would mean that the dataset is not balanced with regards to the diversity of target words. A common way to model the diversity of a distribution of words is to compute its entropy, so we computed the the Pearson correlation between the entropy of the test set's target word distribution in each language and mT5 and ByT5 Acc@3. The entropy of a distribution of word is maximal if every word is different, and it is minimal if it consist on a single word. mT5 and ByT5 displayed correlation coefficients of $-0.16$ and $0.13$ respectively with $p$-value of $0.45$ and $0.53$ respectively. These results show that there is insufficient evidence to conclude that there is a significant linear relationship between target words diversity and model performance because the $p$-value is far above the $0.05$ significance threshold. Hence, target word diversity is not a good predictor of model performance variability across languages.

**Do some languages yield higher scores because they have a smaller vocabulary in their corpora?** We expect that languages with smaller corpora will be easier to fine-tune on because of a

smaller prediction space. To test that hypothesis, we computed the Pearson correlation between test set's vocabulary size and mT5 and ByT5's Acc@3 for each language. mT5 and ByT5 displayed correlation coefficients of $-0.29$ and $0.13$ respectively with $p$-value of $0.17$ and $0.54$ respectively. Thus there is insufficient evidence to conclude that there is a significant linear relationship between vocabulary size and model performance because the $p$-value is above the $0.05$ significance threshold.

**Does similarity to the Latin alphabet impact models' performance?** We verify this hypothesis quantitatively by computing the similarity between a) a Latin alphabet composed of the 26 letters of the alphabet in lower and upper case and b) the alphabet of each language corresponding to all the characters in the test set except punctuation and special characters. The similarity was computed with the Jaccard similarity coefficient (Jaccard, 1908), i.e. the ratio of number of unique items in the intersection of both alphabets and the number of unique items in the union of both alphabets. Moreover we used the same methodology as before and computed the Pearson correlation between the Jaccard similarity index and chrF as this metric is more granular in models' character level performance. We observed a correlation of $0.56$ and $0.75$ for mT5 and ByT5 respectively with $p$-values $< 0.01$ respectively. It indicates that the similarity between the Latin alphabet and each language alphabet is significantly correlated to mT5 and ByT5 chrF.

### D.3.8 Evaluation and Discussion

Whether we used a word level metric like Acc@3 or a character level metric like chrF, ByT5 is more accurate at autocomplete than mT5. We also observe that these models generalize more easily to languages written in an alphabet closer to the Latin alphabet, ByT5 being more sensitive to the alphabet of the input language.

### D.4 Transliteration

### D.4.1 Task description

Transliteration is the conversion of text in one writing system to another writing system, e.g., text written in the Devanagari script to the Latin script. It differs from translation in that it does not change the language content of the text, just the script. Many languages are written in multiple scripts, and the current task involves transliterating whole sentences, not just isolated terms, from one script to another.

### D.4.2 Data Creation and Annotation process

Most of the data for the task comes from the romanized full-string subset of the Dakshina dataset (Roark et al., 2020), in which 10,000 Wikipedia sentences written in the native scripts of the 12 languages were human-romanized by native speakers, resulting in parallel sentences in the native and Latin scripts.[19] Two 10,000 sentence additions were made to this data for the current transliteration task: Amharic Wikipedia sentences were similarly manually romanized by native speakers; and the Punjabi sentences from the Dakshina dataset, originally written in the Gurmukhi (Brahmic) script, were manually transliterated by native speakers to the Shahmukhi (Perso-Arabic) script.

### D.4.3 Data Preparation

The resulting collection allows for overall 30 tasks converting between various scripts. These are summarised in Table 10 where, for each language indicated by the BCP-47 code (Phillips and Davis, 2009), the corresponding transliteration tasks are shown for scripts indicated by their ISO-15924 codes (ISO, 2004).

All the native script data was normalized using Unicode NFC (Whistler, 2021). The data was then further transformed using language-specific visual normalization for Brahmic and Perso-Arabic writing systems using the Nisaba script normalization library (Johny et al., 2021; Gutkin et al., 2022). Both NFC and visual normalization operations preserve visual invariance of the input text, with visual normalization handling many ambiguous cases that fall outside the scope of standard NFC.

### D.4.4 Data Statistics

For each task, we establish 2,000 training sentences, 2,000 development set sentences, and close to 6,000 test sentences. Training data for any pretrained models used in the task cannot include the Dakshina dataset. Since this is a contextual few-shot transliteration benchmark, we do not provide the romanization lexicons that were released in the Dakshina dataset along with the full sentence romanizations.

Our few-shot contextual transliteration task covers 13 languages from 3 language families

---

[19]In the Dakshina distribution, the parallel sentences can be found in files named `LANG.romanized.rejoined.tsv`, where `LANG` is a BCP-47 language code.

(Indo-Aryan, Dravidian and Semitic), all but one (Amharic) from South Asia.

### D.4.5 Directionality and Evaluation Ambiguity

One difference between romanization in these languages and transliteration in the opposite direction (from the Latin script to the native script) is that none of the languages in the benchmark have an orthography in the Latin script, i.e., there is no single correct spelling in the Latin script for these languages. Rather, individuals tend to provide a rough phonetic transcription of the sentences using the Latin script. As a result, word identity may be difficult to achieve (hence high word-error rate), but string similarity should be relatively high between quality romanizations hence we use character-error rate to evaluate the transliterations. The ability to produce romanizations automatically has several key use cases, including simulation of parallel data from mono-script language samples, and for multilingual modeling of languages that use different scripts. For that reason, we include both directions in the benchmark.

### D.4.6 Experimental Setup

Previously Xue et al. (2022) performed ByT5 fine-tuning and evaluation of transliteration and romanization directions separately on single-word, rather than full-sentence, data from vanilla Dakshina dataset. In this benchmark we remove the separation into transliteration and romanization by requiring all tasks to be fine-tuned jointly. In order to achieve this, during all stages of training, development and testing a special code is prepended to the input feature strings for each task. This task code indicates that the input features correspond to the conversion from writing system Source to writing system Target for a language lang. It is encoded as a string "lang_Source_Target". For example, for Punjabi (pa) conversion from Shahmukhi (Arab) to Gurmukhi (Guru) writing systems, the task code is "pa_Arab_Guru".

In the *full* fine-tuning setup[20] we jointly fine-tune the 30 transliteration tasks using mT5 and ByT5 models in Small, Base and Large configurations that correspond to around 300M, 582M and 1.2B parameters, respectively (Xue et al., 2021, 2022). Fine-tuning uses 10K training steps with a batch

---

[20]This *full* fine-tuning setup contrasts with an efficient fine-tuning setup relying on fewer fine-tuning steps and smaller batch sizes, the results of which are reported in Table 4.

size of 128. We used Google TPU-v3 accelerators (Kumar et al., 2019) for fine-tuning all the configurations apart from ByT5 Large for which a more powerful TPU-v4 (Pope et al., 2022) was necessary.

### D.4.7 Evaluation and Discussion

The evaluation results of the full fine-tuning setup described above are provided in Table 11, which shows the character error rate (CER) for each of the 30 transliteration tasks in six configurations, along with the corresponding averages over all the tasks.

Some general trends are observable in these baseline results. The ByT5 error rates are generally substantially better than mT5, and, while the size of the configuration matters for mT5, it does not seem to matter much for ByT5. Overall, romanization is harder, i.e., transliterating into the Latin script yields higher error rates than transliterating out of it, perhaps due to the fact that there is no set orthography in the Latin script in those languages. For the best performing configuration (ByT5-Base), 9 out of 10 of the tasks with the lowest CER are from Latin script to native script. All of the tasks with the highest CER are into either the Latin or Perso-Arabic scripts, and all of the tasks transliterating Perso-Arabic input have worse-than-median CER. In other words: Perso-Arabic input is hard; Latin output is hard; and Perso-Arabic to Latin is particularly hard.

### D.5 Machine Translation

### D.5.1 Data Card

**Basic Info**

1. Original datset name: FLORES-101

2. Repository: https://github.com/facebookresearch/flores/tree/main/flores200

3. Paper: Goyal et al. (2022)

4. Point of Contact (original version): NLLB Team (flores@fb.com)

**Why is this dataset part of XTREME-UP?** Machine translation is an important tool for expanding language coverage for natural language processing tools. FLORES-101 is a high-quality, highly-multilingual dataset.

**Data Fields**

1. input: the source sentence, which is always English (string)

2. target: the target-language translation of the source sentence (string)

**Data Example** {"input": "<2xh> Local media reports an airport fire vehicle rolled over while responding.", "target": "Oonondaba basekuhlaleni bxele ukuba isithuthi somlilo sesitishi senqwelomoya siye saphethuka sisazama ukunceda."}

**Languages** Included in XTREME-UP release (93): Afrikaans (af), Amharic (am), Arabic (ar), (Eastern) Armenian (hy), Assamese (as), (North) Azerbaijani (az), Belarusian (be), Bengali (bn), Bosnian (bs), Bulgarian (bg), Burmese (my), Catalan (ca), Cebuano (ceb), Central Kurdish (ckb), Chinese (zh), Croatian (hr), Czech (cs), Danish (da), Dutch (nl), Estonian (et), Finnish (fi), French (fr), Fula (ff), Galician (gl), Georgian (ka), German (de), Greek (el), Gujarati (gu), Hausa (ha), Hebrew (he), Hindi (hi), Hungarian (hu), Icelandic (is), Igbo (ig), Indonesian (id), Irish (ga), Italian (it), Japanese (ja), Javanese (jv), Kannada (kn), Kazakh (kk), Khmer (km), Korean (ko), Kyrgyz (ky), Lao (lo), Latvian (lv), Lingala (ln), Lithuanian (lt), (Lu)Ganda (lg), Luxembourgish (lb), Macedonian (mk), Malay (ms), Malayalam (ml), Maltese (ml), Maori (mi), Marathi (mr), Mongolian (mn), Nepali (ne), Pedi (Sepedi) (Northern Sotho) (nso), Norwegian (no), Nyanja (Chichewa) (ny), Oriya (or), Oromo (om), Pashto (ps), Persian (fa), Polish (pl), Portuguese (pt), Punjabi (pa), Romanian (ro), Russian (ru), Serbian (sr), Shona (sn), Sindhi (sd), Slovak (sk), Slovenian (sl), Somali (so), Spanish (es), Swahili (sw), Swedish (sv), Tagalog (tl), Tajik (tg), Tamil (ta), Telugu (te), Thai (th), Turkish (tr), Ukrainian (uk), Urdu (ur), Uzbek (uz), Vietnamese (vi), Welsh (cy), Xhosa (xh), Yoruba (yo), Zulu (zu).

Evaluated in benchmark (39): Amharic (am), (Eastern) Armenian (hy), Assamese (as), (North) Azerbaijani (az), Burmese (my), Central Kurdish (ckb), Gujarati (gu), Hausa (ha), Icelandic (is), Igbo (ig), Irish (ga), Javanese (jv), Kannada (kn), Khmer (km), Kyrgyz (ky), Lao (lo), Lingala (ln), (Lu)Ganda (lg), Luxembourgish (lb), Macedonian (mk), Malayalam (ml), Mongolian (mn), Nepali (ne), Pedi (Sepedi) (Northern Sotho) (nso), Nyanja

(Chichewa) (ny), Oromo (om), Pashto (ps), Punjabi (pa), Shona (sn), Sindhi (sd), Somali (so), Swahili (sw), Tajik (tg), Telugu (te), Welsh (cy), Xhosa (xh), Yoruba (yo), Zulu (zu).

**Data Statistics** 50% of the FLORES-101 dev split was reserved for training and the remainder for validation. The original devtest split was unchanged and reserved for testing. This results in 499/498/1012 sentence pairs for train/validation/test, respectively.

**Dataset Curators** The original dataset was curated by the NLLB (No Language Left Behind) Team (flores@fb.com). The version included in XTREME-UP was curated by Parker Riley (prkriley@google.com) and Isaac Caswell (icaswell@google.com).

**Curation Rationale** The original FLORES-101 dataset was created to be able to evaluate machine translation models in many languages. The version released in XTREME-UP was created to focus on low-resource languages and provide an in-domain train split along with validation and test splits, all of sizes in line with other tasks in XTREME-UP.

**Data Sources** The source data (selected by the NLLB Team) comes from Wikinews, Wikijunior, and Wikivoyage.

**Dataset Creation** Details of the creation of the original dataset are available in the original publication (Goyal et al., 2022).

**Changes to the Original Dataset for XTREME-UP** The version of the dataset in XTREME-UP only has the source and target strings, removing additional metadata. We also include 93 of the original 100 non-English languages (the subset supported by Google Translate). Of these, only 39 are used for official evaluation.

## D.6 Question Answering and Retrieval

### D.6.1 Data Card

**Basic Info**

1. Original datset names: TyDi QA, XOR-TyDi QA

2. Additional cross-lingual data was collected as part of XTREME-UP, following similar methodology

**Why is this dataset part of XTREME-UP?** Question answering enables information access.

**Data Fields**

1. question: a question in the target language (string)

2. title: the title of the evidence passage — target language for in-language setting, English for cross-language setting (string)

3. passage: the evidence passage, which might contain an answer to the question — target language for in-language setting, English for cross-language setting (string)

4. answer: the answer (if any) to the question (string)

**Data Example**   See Table 6.

**Languages**   See Table 5.

**Data Statistics**   See Table 1.

**Data Sources**   Evidence text was sourced from Wikipedia.

**Dataset Creation**   Details of the creation of the original dataset are available in the original TyDi QA and XOR QA publications.

### D.7   Named Entity Recognition (NER)

**Dataset and task description**   The dataset contains processed data from MasakhaNER (Adelani et al., 2021) and MasakhaNER 2.0 (Adelani et al., 2022). Both datasets were created by Masakhane[21].

**Why is this dataset part of XTREME-UP?**
Named entity recognition is a fundamental task in natural language processing. The MasakhaNER datasets are high-quality multilingual datasets that provide data in 20 African languages. The data is human-annotated and thus higher quality than automatically collected NER datasets.

**Languages and ISO 639-3 codes**   Bambara (bam), Ghomálá' (bbj), Éwé (ewe), Igbo (ibo), Kinyarwanda (kin), Luganda (lug), Luo (luo), Mossi (mos), Naija (pcm), Chichewa (nya), chiShona (sna), Kiswahili (swa), Setswana (tsn), Akan/Twi (twi), Wolof (wol), isiXhosa (xho), Yorùbá (yor), isiZulu (zul)

---

[21]https://www.masakhane.io/

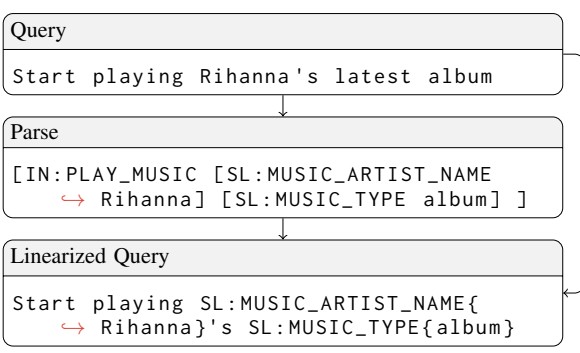

Figure 2: Creation of a linearized query from the actual query and its parse for semantic parsing.

**Changes to the original datasets for XTREME-UP**   The original MasakhaNER datasets are provided in CoNLL format where each input sentence is already tokenized. This makes it difficult to evaluate NER models on natural text where tokenization may often be messy and introduces a bias towards word and subword-based models. To provide a level playing field and to enable evaluation of NER models on natural data data, we process the data in order to align the token-level annotations with byte-level spans in the original pre-tokenized text. For the NER task, we provide the original pre-tokenized text as input to the model. Hausa and Fon subsets of the original data were excluded as matching with the unlabeled source data revealed annotation artefacts in both language subsets.

### D.8   Semantic parsing

#### D.8.1   Task description

Semantic parsing is the task of mapping a natural language utterance to a logical form or a structured interpretation that can be executed by a system such as a virtual assistant. For XTREME-UP, we adapted the MTOP (Li et al., 2021) test dataset to 15 languages, and to 3 code-switched Indic languages. The original MTOP data was published by Facebook and covers 6 languages across 11 domains, 117 intents and 78 slots.

#### D.8.2   Data creation

In this section, we describe the two processes used to extend the MTOP instances: the first involves translation and localization with professional translators and the second code-switching using a language model and verification by human annotators.

In both processes, we perform a linearization step of the query and parse. Given an English utterance from the MTOP English test set and the

corresponding slot information (slot names each with start and end bytes), we add slot tags around corresponding tokens in the query (Figure 2).

**Motivation** Recently, researchers published more multilingual semantic parsing datasets that focus on virtual assistant domains (Li et al., 2021; FitzGerald et al., 2022; Moghe et al., 2022; Goel et al., 2023). We extend a portion of an existing semantic parsing dataset to new languages targeting the following features: a) high-quality utterances produced by professional translators; b) a wide range of domains and intents; c) inclusion of different language families and some underrepresented languages; d) sentences with culturally relevant entities; and e) code-mixed sentences, i.e., multiple language within the same sentence—a common phenomenon in multilingual societies.

**Translating MTOP to 15 languages:** We take the bracketed versions of the slot-tagged English sentences from MTOP and we create translations and localization tasks to be carried out by professional translators. We ran two pilots on a small sample of the data to gather feedback and improve the annotation guidelines. The translators had to translate the original utterances to a given target language, while keeping the brackets around slot value translations and localizing those where possible. Once the pilots were completed without issues, we scaled the tasks to the full test set.

We carried out manual inspections on samples of the data to check if translation and localization was happening correctly, and a set of automatic checks on the full data to ensure that slots were matching between original and translated utterances. Data was sent back to annotators until all the issues were fixed.

**Code-switching MTOP to 3 Indic languages:** We use PaLM to convert the linearized query into a code-mixed query using few-shot prompting. We experimented with different discrete prompt design strategies and selected the best prompts after a qualitative evaluation on a small held-out set (11 examples) covering all 11 domains. Specifically we experimented with three designs.

- **Naive prompting**. The prompt contains (a) the task description followed by a set of examples consisting of (b) the original English linearized query and (c) the corresponding code-mixed version.

- **Parallel sentence prompting**. In this case, the prompt contains (a) the task description, (b) the original English linearized query, and also (c) the target translated query (obtained with Google translate) and (d) the corresponding code-mixed query.

- **Parallel reordered sentence prompting**. Similar to the previous, however, target translated queries are human written.

We observed that the **Parallel sentence prompting** was producing higher quality utterances, with 7/11 correct conversions for Hindi-English. 6/11 for Bengali-English, and 8/11 for Tamil-English. We used this strategy to design prompts with the help of native speakers of those languages. We selected 21 sentences from the training split for creating corresponding exemplars for the prompts. With the latter, we performed few-shot prompting with the 64b PaLM model and converted the test split of MTOP to a code-switched corpora.

Human annotators then had to check the PaLM generated data for the presence of code-mixing and for the labeling to be consistent between the original query and the code-mixed version. The annotators were instructed to fix the automatically generated data whenever they found such issues.

### D.8.3 Data structure and statistics

To create the training, validation and testing splits for MTOP, we start from the English test set and remove intents with less than 10 examples. This leaves us with 53 intents and a maximum of 4,223 examples for each language (some original MTOP languages may have less examples, while our code-switched data may have more due to multiple paraphrases).

For each intent, we randomly select training examples such that each slot is covered by at least one example, for a minimum of 5 examples. We end up with training, development and test sets containing respectively a maximum of 285, 239, and 3,669 instances for each language.

### D.8.4 Experiments

We fine-tune mT5 (Xue et al., 2021) and ByT5 (Xue et al., 2022) in their base and large configurations on the multilingual training data we collected. Table 12 contains the Exact Match accuracies of a multilingual model trained on data from all languages but the code-switched sets. Table 13 contains the results of a model that includes the code-

switched sets. From both tables, we can see that ByT5-base is more accurate then the other models, even compared with the larger ones. This surprising result confirms similar findings on word-level tasks reported by Xue et al. (2022) and Nicosia and Piccinno (2022). We expect mT5 to catch up with ByT5 at larger sizes.

## E    In-context learning examples

We show in-context learning examples for a selection of tasks in Table 14. Each example consists of a general instruction and prefixes for the input and target, which are repeated for each exemplar.

| Language | ISO code | Language family | # of datasets | Resource level | QA | Retrieval | NER | Semantic parsing | MT | ASR | OCR | Transliteration | Auto-complete |
|---|---|---|---|---|---|---|---|---|---|---|---|---|---|
| Afrikaans | af | Indo-European | 2 | 3 | | | | | ✓ | ✓ | | | |
| Amharic | am | Afro-Asiatic | 7 | 2 | | | ✓ | ✓ | ✓ | ✓ | ✓ | ✓ | |
| Assamese | as | Indo-European | 3 | 1 | | | | | ✓ | ✓ | | | |
| Asturian | ast | Indo-European | 2 | 1 | | | | | ✓ | ✓ | | | |
| Azerbaijani | az | Turkic | 3 | 1 | | | | | ✓ | ✓ | | | |
| Ghomálá' | bbj | Atlantic-Congo | 1 | 0 | | | ✓ | | | | | | |
| Belarusian | be | Indo-European | 4 | 3 | | | | ✓ | ✓ | ✓ | | | ✓ |
| Bulgarian | bg | Indo-European | 3 | 3 | | | | | ✓ | ✓ | | | ✓ |
| Bambara | bm | Mande | 1 | 1 | | | ✓ | | | | | | |
| Bengali | bn | Indo-European | 7 | 3 | ✓ | ✓ | | ✓ | ✓ | ✓ | ✓ | ✓ | |
| Bosnian | bs | Indo-European | 2 | 3 | | | | | ✓ | ✓ | | | |
| Cebuano | ceb | Austronesian | 2 | 3 | | | | | ✓ | ✓ | | | |
| Central Kurdish | ckb | Indo-European | 3 | 1 | | | | | ✓ | ✓ | | | |
| Welsh | cy | Indo-European | 3 | 1 | | | | | ✓ | ✓ | | | |
| Danish | da | Indo-European | 3 | 3 | | | | | ✓ | ✓ | | | ✓ |
| Ewe | ee | Atlantic-Congo | 1 | 1 | | | ✓ | | | | | | |
| Greek | el | Indo-European | 3 | 3 | | | | | ✓ | ✓ | | | ✓ |
| Estonian | et | Uralic | 3 | 3 | | | | | ✓ | ✓ | | | ✓ |
| Fula | ff | Atlantic-Congo | 2 | 1 | | | | | ✓ | ✓ | | | |
| Filipino | fil | Austronesian | 2 | 1 | | | | | ✓ | ✓ | | | |
| Irish | ga | Indo-European | 4 | 2 | | | | | ✓ | ✓ | | | ✓ |
| Galician | gl | Indo-European | 3 | 3 | | | | | ✓ | ✓ | | | ✓ |
| Gujarati | gu | Indo-European | 4 | 1 | | | | | ✓ | ✓ | | ✓ | |
| Hausa | ha | Afro-Asiatic | 5 | 2 | | | ✓ | ✓ | ✓ | ✓ | | | |
| Hebrew | he | Afro-Asiatic | 3 | 3 | | | | | ✓ | ✓ | | | ✓ |
| Armenian | hy | Indo-European | 4 | 1 | | | | | ✓ | ✓ | | | ✓ |
| Indonesian | id | Austronesian | 5 | 3 | ✓ | ✓ | | | ✓ | ✓ | | | ✓ |
| Igbo | ig | Atlantic-Congo | 4 | 1 | | | ✓ | | ✓ | ✓ | | | |
| Icelandic | is | Indo-European | 4 | 2 | | | | | ✓ | ✓ | | | ✓ |
| Javanese | jv | Austronesian | 3 | 1 | | | | | ✓ | ✓ | | | |
| Georgian | ka | Kartvelian | 2 | 3 | | | | | ✓ | ✓ | | | |
| Kamba | kam | Atlantic-Congo | 2 | 0 | | | | | ✓ | ✓ | | | |
| Kabuverdianu | kea | Indo-European | 2 | 0 | | | | | ✓ | ✓ | | | |
| Kazakh | kk | Turkic | 2 | 3 | | | | | ✓ | ✓ | | | |
| Khmer | km | Austroasiatic | 3 | 1 | | | | | ✓ | ✓ | | | |
| Kannada | kn | Dravidian | 5 | 1 | | | | | ✓ | ✓ | ✓ | ✓ | |
| Kyrgyz | ky | Turkic | 3 | 1 | | | | | ✓ | ✓ | | | |
| Luxembourgish | lb | Indo-European | 3 | 1 | | | | | ✓ | ✓ | | | |
| (Lu)Ganda | lg | Atlantic-Congo | 4 | 1 | | | ✓ | | ✓ | ✓ | | | |
| Lingala | ln | Atlantic-Congo | 3 | 1 | | | | | ✓ | ✓ | | | |
| Lao | lo | Tai-Kadai | 3 | 2 | | | | | ✓ | ✓ | | | |
| Lithuanian | lt | Indo-European | 3 | 3 | | | | | ✓ | ✓ | | | ✓ |
| (Dho)Luo | luo | Nilotic | 3 | 0 | | | ✓ | | ✓ | ✓ | | | |
| Latvian | lv | Indo-European | 3 | 3 | | | | | ✓ | ✓ | | | ✓ |
| Macedonian | mk | Indo-European | 3 | 1 | | | | | ✓ | ✓ | | | |
| Malayalam | ml | Dravidian | 4 | 1 | | | | | ✓ | ✓ | | ✓ | |
| Mongolian | mn | Mongolic-Khitan | 3 | 1 | | | | | ✓ | ✓ | | | |
| Mossi (Mooré) | mos | Atlantic-Congo | 1 | 0 | | | ✓ | | | | | | |
| Marathi | mr | Indo-European | 3 | 2 | | | | | ✓ | ✓ | | ✓ | |
| Malay | ms | Austronesian | 2 | 3 | | | | | ✓ | ✓ | | | |
| Maltese | mt | Afro-Asiatic | 2 | 2 | | | | | ✓ | ✓ | | | |
| Burmese | my | Sino-Tibetan | 4 | 1 | | | | | ✓ | ✓ | ✓ | | |
| Nepali | ne | Indo-European | 3 | 1 | | | | | ✓ | ✓ | | | |
| Norwegian | no | Indo-European | 2 | 1 | | | | | ✓ | | | | |
| Northern Sotho | nso | Atlantic-Congo | 3 | 1 | | | | | ✓ | ✓ | | | |
| Nyanja (Chichewa) | ny | Atlantic-Congo | 4 | 1 | | | ✓ | | ✓ | ✓ | | | |
| Occitan | oc | Indo-European | 2 | 1 | | | | | ✓ | ✓ | | | |
| Oromo | om | Afro-Asiatic | 3 | 1 | | | | | ✓ | ✓ | | | |
| Oriya | or | Indo-European | 2 | 1 | | | | | ✓ | ✓ | | | |
| Punjabi | pa | Indo-European | 4 | 2 | | | | | ✓ | ✓ | | ✓ | |
| Nigerian Pidgin | pcm | Indo-European | 2 | 0 | | | ✓ | | | | | | ✓ |
| Pashto | ps | Indo-European | 3 | 1 | | | | | ✓ | ✓ | | | |
| Romanian | ro | Indo-European | 3 | 3 | | | | | ✓ | ✓ | | | ✓ |
| Kinyarwanda | rw | Atlantic-Congo | 1 | 1 | | | ✓ | | | | | | |
| Sanskrit | sa | Indo-European | 1 | 2 | | | | | | | ✓ | | |
| Sindhi | sd | Indo-European | 4 | 1 | | | | | ✓ | ✓ | | ✓ | |
| Sinhala | si | Indo-European | 2 | 0 | | | | | | | ✓ | ✓ | |
| Slovak | sk | Indo-European | 3 | 3 | | | | | ✓ | ✓ | | | ✓ |
| Slovenian | sl | Indo-European | 3 | 3 | | | | | ✓ | ✓ | | | ✓ |
| Shona | sn | Atlantic-Congo | 4 | 1 | | | ✓ | | ✓ | ✓ | | | |
| Somali | so | Afro-Asiatic | 3 | 1 | | | | | ✓ | ✓ | | | |
| Swahili | sw | Atlantic-Congo | 8 | 2 | ✓ | ✓ | ✓ | ✓ | ✓ | ✓ | ✓ | | |
| Tamil | ta | Dravidian | 4 | 3 | | | | ✓ | ✓ | ✓ | | ✓ | |
| Telugu | te | Dravidian | 6 | 1 | ✓ | ✓ | | | ✓ | ✓ | | ✓ | |
| Tajik | tg | Indo-European | 3 | 1 | | | | | ✓ | ✓ | | | |
| Thai | th | Tai-Kadai | 3 | 3 | | | | ✓ | ✓ | ✓ | | | |
| Tswana (Setswana) | tn | Atlantic-Congo | 1 | 2 | | | ✓ | | | | | | |
| Twi | tw | Atlantic-Congo | 1 | 1 | | | ✓ | | | | | | |
| Uyghur | ug | Turkic | 1 | 1 | | | | | | | | | ✓ |
| Ukrainian | uk | Indo-European | 3 | 3 | | | | | ✓ | ✓ | | | ✓ |
| Umbundu | umb | Atlantic-Congo | 2 | 0 | | | | | ✓ | ✓ | | | |
| Urdu | ur | Indo-European | 4 | 3 | | | | | ✓ | ✓ | | ✓ | ✓ |
| Uzbek | uz | Turkic | 2 | 3 | | | | | ✓ | ✓ | | | |
| Wolof | wo | Atlantic-Congo | 3 | 2 | | | ✓ | | ✓ | ✓ | | | |
| Xhosa | xh | Atlantic-Congo | 4 | 2 | | | ✓ | | ✓ | ✓ | | | |
| Yoruba | yo | Atlantic-Congo | 5 | 2 | | | ✓ | ✓ | ✓ | ✓ | | | |
| Zulu | zu | Atlantic-Congo | 5 | 2 | | | ✓ | ✓ | ✓ | ✓ | | | |

Table 5: Overview of under-represented languages covered in XTREME-UP.

| Task | Language | Input | Output |
|---|---|---|---|
| Speech Recognition | Swahili | *[audio waveform]* | marekebisho au maombi yoyote laz-ima yafuatwe kupitia wakala wa kusafiri kwanza na si moja kwa moja na hoteli |
| Document OCR | Burmese | ဆိုလိုက်သည်။၄ရက်ဇ္ဇရ။ ၈နွ ဖြုတပ်ရှ၁၀၀ တွင်း ၵၢ၁ကၢ၊ ဧမ္မဏ်ဝမ်းလှ္ရ၁၀၀၊ အဖြောက်ဖုံးဂရက်ဖျာ ၤၣ်၊ ၵုၣ်းသို့တက်စေၟ၊ မြမ္မ၁တပ်ကၢဆီးတၢ၁ပြစ် ခတ်ရၢ၊ ၾၢပြစ်ၵုၣ်ဖံၣ်။ ၣၣ်မၢၢ်ဝမ်ဆိုင်း မၢကြၢက် | ဆိုလိုက်သည်။၄ရက်ဇ္ဇရ။ ၈နွ ဖြုတပ်ရှ၁၀၀ တွင်း ၵၢ၁ကၢ၊ အမ္မ၀ဒမ်းလှ္ရ၁၀၀၊ အဖြောက်ဖုံးဂရက်ဖျာ ၤၣ်၊ ၵုၣ်းသို့တက်စေၟ၊ မြမ္မ၁တပ်ကၢဆီးတၢ၁မြစ် ခတ်ရၢ၊အၢပြစ်ၵုၣ်ဖံၣ်။ ၣၣ်မရၢ်ဝမ်ဆိုင်း မၢကြၢက် |
| Autocomplete | Nigerian Pidgin | make I just dey go back to my papa hou | house |
| Transliteration | Marathi | surguja bhagatale rahivahi. | सुरगुजा भागातले रहिवासी |
| Machine Translation | Xhosa | It was developed by John Smith in the 1970s to help inexperienced folders or those with limited motor skills. | Yeenziwa nguJohn Smith kwiminyaka yee-1970 ukunceda iifolda ezingena-mava okanye ezo zinobuchule bemoto obulinganiselweyo. |
| In-language Retrieval for QA | Telugu | ఆంధ్రజ్యోతి పత్రిక యజమాని ఎవరు? | `Title:` వేమూరి రాధాకృష్ణ
`Passage:` వేమూరి రాధాకృష్ణ ఆంధ్రజ్యోతి పత్రిక ప్రధాన సంపాదకులు మరియు మేనేజింగ్ డైరెక్టర్. ఏబీఎన్ ఆంధ్రజ్యోతి టీవీ ఛానల్లో ఇతని కార్యక్రమం ఓపెన్ హార్ట్ విత్ ఆర్కే మిక్కిలి ప్రాచుర్యం పొందినది. ఆంధ్రజ్యోతి దినపత్రికలో సాధారణ విలేఖరిగా జీవితాన్ని మొదలుపెట్టి చివరికి దాని యజమాని స్థాయికి ఎదిగారు.[1]. |
| In-language QA | Telugu | `Question:` ఆంధ్రజ్యోతి పత్రిక యజమాని ఎవరు?
`Title:` వేమూరి రాధాకృష్ణ
`Passage:` వేమూరి రాధాకృష్ణ ఆంధ్రజ్యోతి పత్రిక ప్రధాన సంపాదకులు మరియు మేనేజింగ్ డైరెక్టర్. ఏబీఎన్ ఆంధ్రజ్యోతి టీవీ ఛానల్లో ఇతని కార్యక్రమం ఓపెన్ హార్ట్ విత్ ఆర్కే మిక్కిలి ప్రాచుర్యం పొందినది. ఆంధ్రజ్యోతి దినపత్రికలో సాధారణ విలేఖరిగా జీవితాన్ని మొదలుపెట్టి చివరికి దాని యజమాని స్థాయికి ఎదిగారు.[1]. | వేమూరి రాధాకృష్ణ

(or "No Answer" for some examples) |
| Cross-language Retrieval for QA | Oriya | ପ୍ରଶ୍ନ :କିଏ ସତ୍ୟବତୀ ଏମ୍ ଶୀର୍ଷଙ୍କର ଚାପା ଅଟନ୍ତି? | `Title:` Satyavati
`Context:` Daughter of the Chedi king, Vasu (also known as Uparichara Vasu) and a cursed "apsara" (celestial nymph) who was turned into a fish called Adrika, Satyavati was brought up as a commoner... |
| Cross-language QA | Oriya | `Question:` ପ୍ରଶ୍ନ :କିଏ ସତ୍ୟବତୀ ଏମ୍ ଶୀର୍ଷଙ୍କର ଚାପା ଅଟନ୍ତି?
`Title:` Satyavati
`Context:` Daughter of the Chedi king, Vasu (also known as Uparichara Vasu) and a cursed "apsara" (celestial nymph) who was turned into a fish called Adrika, Satyavati was brought up as a commoner... | Uparichara Vasu

(or "No Answer" for some examples) |
| NER | Wolof | Dafa di, nag, Ërob rawatina Farās, dañuy xeex ak a bunduxataal tuu-taafóoni waaso yi. | LOC: Ërob
LOC: Farās |
| Semantic Parsing | Zulu | Ingabe ikhona imicimbi yasendaweni eqhubekayo kuleli sonto | `[IN:GET_EVENT`
`  [SL:ATTRIBUTE_EVENT yasendaweni]`
`  [SL:DATE_TIME kuleli sonto]`
`]` |

Table 6: Examples of each task in XTREME-UP. The tasks are generally text-in, text-out with a few exceptions. On the output side, autocomplete requires generating the top-3 outputs and retrieval outputs document identifiers—current systems tend to implement retrieval by mapping both inputs and candidate outputs to vector and performing nearest neighbor lookup. On the input side, speech recognition has audio input and document OCR has image outputs; our initial baseline systems use external systems to map this to text as a preprocessing step, though we hope to see multi-modal systems eliminate this step in the near future.

| Task Code | Maestro-U | mT5 Base | mT5 Small | ByT5 Base | Task Code | Maestro-U | mT5 Base | mT5 Small | ByT5 Base |
|---|---|---|---|---|---|---|---|---|---|
| af_za | 4.19 | 4.19 | 4.24 | 4.19 | ... | ... continued ... | | | ... |
| am_et | 8.60 | 8.60 | 8.66 | 8.6 | mt_mt | 11.48 | 11.57 | 11.56 | 11.57 |
| ar_eg | 6.00 | 6.00 | 6.03 | 6.00 | my_mm | 14.70 | 14.70 | 14.87 | 14.66 |
| as_in | 8.49 | 8.49 | 8.56 | 8.49 | nb_no | 4.14 | 4.14 | 4.21 | 4.14 |
| ast_es | 4.49 | 4.49 | 4.66 | 4.49 | ne_np | 9.22 | 9.26 | 9.25 | 9.22 |
| az_az | 5.67 | 5.67 | 5.7 | 5.67 | nl_nl | 3.15 | 3.15 | 3.27 | 3.15 |
| be_by | 3.34 | 3.34 | 3.42 | 3.34 | nso_za | 7.13 | 7.13 | 7.17 | 7.13 |
| bn_in | 6.16 | 6.16 | 6.20 | 6.16 | ny_mw | 7.08 | 7.08 | 7.08 | 6.95 |
| bs_ba | 2.93 | 2.93 | 3.05 | 2.93 | oc_fr | 7.68 | 7.68 | 7.84 | 7.68 |
| ca_es | 2.72 | 2.72 | 2.76 | 2.72 | om_et | 14.36 | 14.36 | 14.52 | 14.36 |
| ceb | 4.36 | 4.36 | 4.49 | 4.36 | or_in | 7.42 | 7.42 | 8.70 | 7.42 |
| ckb_iq | 8.70 | 8.70 | 8.77 | 8.70 | pa_in | 7.35 | 7.35 | 7.38 | 7.35 |
| cmn_hans_cn | 16.48 | 16.06 | 16.12 | 16.06 | pl_pl | 2.49 | 2.49 | 2.52 | 2.49 |
| cmn_hant_hk | 34.82 | 34.23 | 34.24 | 34.13 | ps_af | 16.82 | 16.82 | 16.86 | 16.82 |
| cs_cz | 3.30 | 3.30 | 3.35 | 3.30 | pt_br | 2.87 | 2.87 | 3.05 | 2.87 |
| cy_gb | 7.11 | 7.11 | 7.17 | 7.11 | ro_ro | 3.58 | 3.58 | 3.64 | 3.58 |
| da_dk | 6.30 | 6.30 | 6.35 | 6.30 | rup_bg | 2.72 | 2.72 | 2.86 | 2.72 |
| de_de | 2.39 | 2.39 | 2.46 | 2.39 | ru_ru | 3.05 | 2.86 | 3.09 | 2.87 |
| el_gr | 4.73 | 4.73 | 4.77 | 4.73 | sd_arab_in | 9.22 | 9.22 | 9.65 | 10.08 |
| en_us | 9.02 | 9.02 | 9.11 | 9.02 | sk_sk | 2.39 | 2.39 | 2.43 | 2.39 |
| es_419 | 1.81 | 1.81 | 1.85 | 1.81 | sl_si | 4.58 | 4.58 | 4.60 | 4.17 |
| et_ee | 2.24 | 2.24 | 2.28 | 2.24 | sn_zw | 9.45 | 9.45 | 9.48 | 9.45 |
| fa_ir | 4.96 | 4.96 | 5.87 | 4.96 | so_so | 13.73 | 13.73 | 13.81 | 13.73 |
| ff_sn | 21.22 | 21.22 | 21.42 | 21.22 | sr_rs | 9.93 | 9.93 | 9.95 | 9.95 |
| fi_fi | 2.02 | 2.02 | 2.05 | 2.02 | sv_se | 4.21 | 4.21 | 4.30 | 4.21 |
| fil_ph | 3.59 | 3.59 | 3.63 | 3.59 | sw_ke | 12.62 | 12.62 | 12.76 | 12.62 |
| fr_fr | 4.57 | 4.57 | 4.63 | 4.57 | ta_in | 12.35 | 11.55 | 15.06 | 12.35 |
| ga_ie | 29.75 | 29.75 | 29.78 | 29.79 | te_in | 7.48 | 7.48 | 7.56 | 7.48 |
| gl_es | 2.55 | 2.55 | 2.58 | 2.55 | tg_tj | 4.56 | 4.56 | 4.60 | 4.56 |
| gu_in | 5.75 | 5.75 | 5.9 | 5.75 | th_th | 11.89 | 11.89 | 11.92 | 11.48 |
| ha_ng | 7.70 | 7.70 | 9.23 | 6.90 | tr_tr | 4.28 | 4.28 | 4.34 | 4.28 |
| he_il | 18.36 | 18.36 | 18.40 | 18.36 | uk_ua | 5.44 | 5.44 | 5.47 | 5.44 |
| hi_in | 5.59 | 5.59 | 5.63 | 5.59 | umb_ao | 17.46 | 17.09 | 17.47 | 14.98 |
| hr_hr | 4.46 | 4.46 | 4.56 | 4.46 | ur_pk | 7.61 | 7.61 | 7.63 | 7.61 |
| hu_hu | 7.05 | 7.05 | 7.10 | 7.05 | uz_uz | 7.40 | 7.40 | 7.42 | 7.40 |
| hy_am | 4.93 | 6.25 | 4.94 | 4.93 | vi_vn | 11.80 | 11.80 | 11.83 | 11.80 |
| id_id | 3.14 | 3.14 | 3.16 | 3.14 | wo_sn | 15.26 | 15.26 | 15.29 | 15.26 |
| ig_ng | 14.06 | 14.06 | 14.29 | 14.07 | xh_za | 16.65 | 16.65 | 16.69 | 16.68 |
| is_is | 6.23 | 6.23 | 6.25 | 6.23 | yo_ng | 19.84 | 19.84 | 19.93 | 19.84 |
| it_it | 1.39 | 1.39 | 1.44 | 1.39 | zu_za | 5.56 | 5.56 | 5.62 | 5.56 |
| ja_jp | 25.74 | 25.49 | 25.51 | 25.43 | Micro-Average | 8.28 | 8.27 | 8.40 | 8.22 |
| jv_id | 4.66 | 4.66 | 4.72 | 4.52 | | | | | |
| ka_ge | 10.09 | 10.09 | 10.16 | 10.09 | | | | | |
| kam_ke | 11.74 | 11.69 | 11.78 | 11.74 | | | | | |
| kea_cv | 4.11 | 4.11 | 4.17 | 4.11 | | | | | |
| kk_kz | 3.58 | 3.58 | 3.66 | 3.58 | | | | | |
| km_kh | 20.15 | 20.15 | 20.15 | 20.15 | | | | | |
| kn_in | 5.13 | 5.13 | 5.40 | 5.13 | | | | | |
| ko_kr | 14.29 | 14.29 | 14.22 | 14.29 | | | | | |
| ky_kg | 4.53 | 4.53 | 4.56 | 4.44 | | | | | |
| lb_lu | 13.54 | 13.54 | 13.64 | 13.54 | | | | | |
| lg_ug | 8.99 | 8.99 | 9.13 | 8.99 | | | | | |
| ln_cd | 4.61 | 4.61 | 4.76 | 4.61 | | | | | |
| lo_la | 22.80 | 22.80 | 22.84 | 23.25 | | | | | |
| lt_lt | 4.51 | 4.51 | 4.55 | 4.51 | | | | | |
| luo_ke | 5.64 | 5.64 | 5.73 | 5.64 | | | | | |
| lv_lv | 2.18 | 2.18 | 2.21 | 2.18 | | | | | |
| mi_nz | 9.59 | 9.51 | 9.6 | 8.68 | | | | | |
| mk_mk | 3.60 | 3.60 | 3.66 | 3.60 | | | | | |
| ml_in | 5.04 | 5.45 | 5.2 | 5.07 | | | | | |
| mn_mn | 8.43 | 8.43 | 8.46 | 8.43 | | | | | |
| mr_in | 7.37 | 7.37 | 7.48 | 7.37 | | | | | |
| ms_my | 3.89 | 3.89 | 3.92 | 3.89 | | | | | |

Table 7: ASR tasks evaluated using CER metric at 4K steps of fine-tuning mT5 and ByT5 Small and Base models.

| Training set | | | |
|---|---|---|---|
| | min | mean | max |
| context length | 1 | 28 | 96 |
| target length | 5 | 8 | 32 |
| Validation set | | | |
| | min | mean | max |
| context length | 1 | 28 | 86 |
| target length | 5 | 8 | 23 |
| Test set | | | |
| | min | mean | max |
| context length | 1 | 28 | 86 |
| target length | 5 | 8 | 28 |

Table 8: Context and target character length statistics averaged over the 23 languages of autocomplete.

| | Acc@3 | | chrF | |
|---|---|---|---|---|
| Language | mT5 | ByT5 | mT5 | ByT5 |
| be | 12.4 | 8.99 | 22.12 | 26.02 |
| bg | 14.45 | 15.38 | 23.27 | 29.92 |
| da | 13.36 | 26.95 | 22.33 | 35.03 |
| el | 21.74 | 14.91 | 25.3 | 32.81 |
| en | 19.2 | 40.1 | 23.08 | 39.84 |
| et | 8 | 23.2 | 20.3 | 31.85 |
| eu | 9.7 | 22 | 26.36 | 34.52 |
| ga | 11.8 | 23.06 | 22.08 | 30.66 |
| gd | 26.28 | 31.85 | 32.79 | 37.65 |
| gl | 19 | 41.4 | 26.53 | 45.76 |
| he | 8.65 | 13.87 | 18.76 | 23.25 |
| hy | 3.49 | 6.98 | 14.98 | 24.4 |
| id | 18.59 | 37.92 | 28.67 | 41.34 |
| is | 26.37 | 32.45 | 27.96 | 32.39 |
| lt | 8.41 | 21.03 | 21.44 | 31.39 |
| lv | 9.6 | 19 | 22.72 | 31.87 |
| pcm | 30.3 | 36.42 | 31.32 | 39 |
| ro | 11.9 | 21.4 | 22.79 | 32.39 |
| sk | 17.49 | 26.23 | 24.56 | 33.75 |
| sl | 14.1 | 25.6 | 23.54 | 33.41 |
| ug | 3.41 | 0.23 | 15.72 | 14.88 |
| uk | 11.3 | 13.04 | 23.36 | 29.78 |
| ur | 21.96 | 23.33 | 23.57 | 29.22 |
| **Average** | **14.85** | **22.84** | **23.63** | **32.22** |

Table 9: mT5 and ByT5 performance averaged on the 23 languages test sets after 10 epochs of fine-tuning on ablation dataset.

| Lang. | Tasks | Lang. | Tasks |
|---|---|---|---|
| am | Ethi↔Latn | | Guru↔Latn |
| bn | Beng↔Latn | pa | Arab↔Latn |
| gu | Gujr↔Latn | | Guru↔Arab |
| hi | Deva↔Latn | sd | Arab↔Latn |
| kn | Knda↔Latn | si | Sinh↔Latn |
| ml | Mlym↔Latn | ta | Taml↔Latn |
| mr | Deva↔Latn | te | Telu↔Latn |
| | | ur | Arab↔Latn |

Table 10: Summary of the transliteration tasks.

| Task Code | mT5 | | | ByT5 | | |
|---|---|---|---|---|---|---|
| | Small | Base | Large | Small | Base | Large |
| am_Ethi_Latn | 16.72 | 16.20 | 15.77 | 10.80 | 11.61 | 11.85 |
| am_Latn_Ethi | 17.61 | 14.25 | 13.08 | 8.58 | 8.19 | 8.94 |
| bn_Beng_Latn | 14.94 | 12.78 | 12.06 | 8.58 | 8.70 | 8.88 |
| bn_Latn_Beng | 17.07 | 12.90 | 11.26 | 5.67 | 5.44 | 6.15 |
| gu_Gujr_Latn | 15.58 | 14.19 | 14.31 | 11.79 | 12.37 | 12.14 |
| gu_Latn_Gujr | 10.49 | 7.92 | 6.94 | 2.86 | 2.83 | 3.21 |
| hi_Deva_Latn | 10.59 | 9.63 | 9.49 | 7.24 | 7.71 | 7.70 |
| hi_Latn_Deva | 12.09 | 9.11 | 8.08 | 5.00 | 4.57 | 4.98 |
| kn_Knda_Latn | 11.35 | 8.54 | 8.05 | 4.37 | 4.80 | 4.55 |
| kn_Latn_Knda | 14.28 | 10.95 | 9.57 | 3.72 | 3.62 | 3.88 |
| ml_Latn_Mlym | 21.10 | 16.41 | 14.25 | 5.66 | 5.31 | 5.95 |
| ml_Mlym_Latn | 15.54 | 13.27 | 12.29 | 8.18 | 8.43 | 8.46 |
| mr_Deva_Latn | 11.40 | 9.70 | 9.25 | 5.77 | 6.04 | 6.07 |
| mr_Latn_Deva | 13.68 | 10.92 | 9.58 | 4.94 | 4.73 | 5.40 |
| pa_Arab_Guru | 13.53 | 11.49 | 10.99 | 7.67 | 7.72 | 8.14 |
| pa_Arab_Latn | 21.89 | 19.62 | 19.44 | 16.03 | 16.03 | 16.59 |
| pa_Guru_Arab | 14.26 | 14.10 | 13.82 | 10.76 | 11.06 | 11.32 |
| pa_Guru_Latn | 15.81 | 15.21 | 15.13 | 12.18 | 12.36 | 12.78 |
| pa_Latn_Arab | 19.55 | 17.91 | 16.96 | 12.90 | 13.06 | 13.77 |
| pa_Latn_Guru | 12.78 | 10.38 | 9.42 | 5.47 | 5.11 | 5.72 |
| sd_Arab_Latn | 18.27 | 17.24 | 17.03 | 14.20 | 14.66 | 14.45 |
| sd_Latn_Arab | 14.00 | 9.90 | 8.65 | 9.98 | 5.11 | 6.70 |
| si_Latn_Sinh | 14.79 | 12.34 | 11.31 | 6.06 | 5.78 | 6.24 |
| si_Sinh_Latn | 10.76 | 9.43 | 9.06 | 6.36 | 6.79 | 6.62 |
| ta_Latn_Taml | 18.92 | 14.57 | 12.89 | 5.15 | 4.70 | 5.06 |
| ta_Taml_Latn | 15.83 | 13.01 | 12.58 | 8.35 | 8.70 | 9.04 |
| te_Latn_Telu | 18.16 | 13.60 | 11.70 | 5.01 | 4.82 | 5.27 |
| te_Telu_Latn | 14.50 | 11.26 | 10.51 | 7.25 | 7.30 | 7.31 |
| ur_Arab_Latn | 16.73 | 15.01 | 14.28 | 11.88 | 11.58 | 12.17 |
| ur_Latn_Arab | 13.01 | 10.07 | 8.30 | 5.23 | 5.03 | 5.59 |
| Average | 15.17 | 12.73 | 11.87 | 7.92 | 7.81 | 8.16 |

Table 11: Transliteration tasks evaluated using CER metric at 10K steps of fine-tuning the mT5 and ByT5 models in Small, Base and Large configurations.

| | mT5 | | ByT5 | |
|---|---|---|---|---|
| Language | base | large | base | large |
| am | 20.01 | 26.47 | 33.41 | 25.60 |
| be | 27.82 | 37.52 | 46.72 | 37.36 |
| bn | 29.06 | 37.66 | 45.07 | 35.69 |
| de | 33.71 | 39.96 | 45.34 | 37.93 |
| de (loc) | 33.31 | 40.58 | 45.81 | 38.20 |
| en | 34.09 | 40.39 | 49.50 | 39.52 |
| es | 34.95 | 41.52 | 48.73 | 39.29 |
| fi | 26.63 | 35.74 | 46.80 | 37.17 |
| fr | 33.72 | 40.29 | 48.97 | 39.84 |
| ha | 21.84 | 27.60 | 42.07 | 29.98 |
| hi | 27.89 | 37.59 | 42.26 | 35.42 |
| hu | 25.87 | 33.47 | 43.82 | 35.98 |
| ja | 28.71 | 33.68 | 45.23 | 35.90 |
| pt_br | 33.98 | 39.12 | 47.90 | 39.50 |
| ru | 34.44 | 41.36 | 48.58 | 42.80 |
| sw | 24.06 | 30.25 | 39.96 | 32.09 |
| ta | 25.03 | 33.20 | 43.31 | 31.41 |
| th | 23.81 | 34.35 | 43.80 | 35.30 |
| tr | 27.44 | 36.44 | 44.58 | 36.63 |
| yo | 14.52 | 16.30 | 30.39 | 18.44 |
| zu | 18.73 | 26.79 | 36.96 | 27.49 |
| **Average** | 27.6 | 34.78 | **43.77** | 34.84 |

Table 12: Semantic Parsing: Exact Match (EM) accuracies of mT5 and ByT5 models of different sizes trained multilingually on few-shot data. We report accuracies on all languages.

|  | mT5 | | ByT5 | |
| Language | base | large | base | large |
| --- | --- | --- | --- | --- |
| bn | 10.72 | 11.78 | 22.69 | 16.25 |
| hi | 16.05 | 18.48 | 25.03 | 17.69 |
| ta | 16.98 | 19.71 | 26.21 | 19.27 |

Table 13: Semantic Parsing: Exact Match (EM) accuracies of mT5 and ByT5 models of different sizes trained multilingually on few-shot data. Here the multilingual training data includes three code-switched Indic languages and we report EM for such languages.

| Task | In-context learning example |
| --- | --- |
| Translation | Translate between English and Afrikaans.
English: [INPUT]
Afrikaans: [TARGET] |
| ASR | Correct the ASR output in Afrikaans.
ASR Afrikaans output: [INPUT]
Corrected: [TARGET] |
| NER | Tag the named entities in the Swahili text as person (PER), organization (ORG), location (LOC), and date (DATE). Use $$ as delimiter.
Swahili text: [INPUT]
Named entities: [TARGET] |
| Autocomplete | Complete the Urdu sentence. Write the next word or finish the last word if it is incomplete.
Urdu sentence: [INPUT]
Completion: [TARGET] |

Table 14: In-context learning examples.