# OpenReview forum: "XTREME-UP: A User-Centric Scarce-Data Benchmark for Under-Represented Languages"
_EMNLP/2023/Conference — EMNLP 2023 Findings_

### Official Review · Reviewer_qsF6 · 2023-08-03

**Soundness:** 3

**Excitement:**

3: Ambivalent: It has merits (e.g., it reports state-of-the-art results, the idea is nice), but there are key weaknesses (e.g., it describes incremental work), and it can significantly benefit from another round of revision. However, I won't object to accepting it if my co-reviewers champion it.

**Paper Topic And Main Contributions:**

The paper proposes a benchmark for data-scarce, user-centric and under-represented languages. It evaluates the capability of language models across 88 under-represented languages over 9 tasks. Additionally, several novel datasets have been proposed.

**Questions For The Authors:**

You can see the Reasons To Reject section.

**Reasons To Accept:**

1) Well-written benchmark paper. Benchmark collection rationale and process has been clearly explained. The diagrams and the tables are self explanatory.
2) Important topic of UL languages.
3) New dataset will benefit the community.
4) Benchmark will serve as a good contribution for evaluation of multi-lingual LLMs.

**Reasons To Reject:**

1) The choice and the number of baselines is not justified. I understand when the authors say: "Given that our focus in this paper is on
the dataset and task setup rather than system building, we do not focus on offering novel modeling types nor do we exhaustively evaluate all possible models;". Though I agree that proposing a novel method is beyond scope, but just mT5 and ByT5 is concerning.

2) The experimental setting is a bit flawed in my opinion (from a literature perspective). The setups for ASR is not what is generally used. A lot of the tasks like NER and QA is not generally not treated as a sequence-to-sequence task commonly and more of a token classification task. Again, I am aware of papers that do treat it as a seq-to-seq task, but it is not the traditional method to evaluate these tasks.

3) The analyses section needs more in-depth intuitions. Why do models suffer in African languages? What is the reason for low vs high performance on some languages? Is there a previous reference to the reason atleast?

**Reproducibility:**

4: Could mostly reproduce the results, but there may be some variation because of sample variance or minor variations in their interpretation of the protocol or method.

**Reviewer Confidence:**

5: Positive that my evaluation is correct. I read the paper very carefully and I am very familiar with related work.

---

> ### Author Rebuttal · Authors · 2023-08-28
>
> Thank you for the review.
>
> > 1. The choice and the number of baselines is not justified.
>
> We select three representative methods, each from a different model family: one subword-based method, one byte-based method, and one in-context learning method. We focused on optimizing the experimental settings for the open-source models so that they can be easily reproduced (rather than evaluating a larger number of under-tuned methods).
>
> > 2. The experimental setting is a bit flawed in my opinion (from a literature perspective). The setups for ASR is not what is generally used. A lot of the tasks like NER and QA is not generally not treated as a sequence-to-sequence task commonly and more of a token classification task.
>
> We included multi-modal tasks such as ASR and OCR as they are widely used technologies and are thus important to be made available in a broader set of languages. To make them accessible for the current text-only models and to enable NLP researchers to improve performance for certain languages while staying within the text modality, we provide intermediate outputs of ASR and OCR systems. This post-correction setting has been studied in prior work (e.g., Dutta et al., 2022, https://arxiv.org/abs/2202.01157;  Rijhwani et al., 2020, https://aclanthology.org/2020.emnlp-main.478/). Nevertheless, we expect multi-modal models to be available soon that can tackle these and the other tasks in UP in an end-to-end fashion.
>
> NER and QA have been treated as token classification tasks with the previous generation of encoder-only language models such as BERT. With the recent generation of decoder-only and encoder-decoder generative LLMs, a sequence-to-sequence encoding has become much more commonplace, even for traditional NLP tasks such as NER. We highlight a few studies here that study these tasks in a seq2seq setting. For QA: Khahabi et al. (2020; https://aclanthology.org/2020.findings-emnlp.171/); Asai et al. (2021; https://arxiv.org/abs/2107.11976). For NER: Yan et al. (2021; https://aclanthology.org/2021.acl-long.451/); Lu et al. (2022; https://aclanthology.org/2022.acl-long.395/); Zhang et al. (2022; https://aclanthology.org/2022.acl-long.59/); Shen et al. (2023; https://aclanthology.org/2023.acl-long.698/). In addition, all studies we are aware of that evaluate recent state-of-the-art models such as GPT-3, ChatGPT, GPT-4, PaLM, PaLM 2, LlaMa, and LlaMa-2 follow a sequence-to-sequence format. For these reasons, we adopt the sequence-to-sequence format as the unified format across tasks. We note that datasets can be easily converted back to their standard formats (e.g., span extraction, sequence labeling) to evaluate encoder-only models if desired.
>
> > 3. The analyses section needs more in-depth intuitions.
>
> We observe a correlation between the amount of data of a language during pre-training and a model’s downstream performance on that language. For instance, there is little data available in African languages in common pre-training sources and performance is consequently low for African languages. Similarly, we observe a correlation with the resource category of a language where performance is lower the lower-resourced a language is overall. Finally, we observe the script to be a major factor (see, for instance, the analysis on the impact of the script on the Auto-complete task in the appendix). In particular, performance is generally higher for languages which employ a Latin script due to additional lexical overlap. We highlight additional studies that perform similar analyses on a more limited set of languages: Rust et al. (2021; https://aclanthology.org/2021.acl-long.243/); Hu et al. (2020; https://arxiv.org/abs/2003.11080); Adelani et al. (2022; https://aclanthology.org/2022.emnlp-main.298/). We will extend the analysis justification and add additional justifications in the revised version of the paper.

---

### Official Review · Reviewer_tT4X · 2023-08-04

**Soundness:** 3

**Excitement:**

3: Ambivalent: It has merits (e.g., it reports state-of-the-art results, the idea is nice), but there are key weaknesses (e.g., it describes incremental work), and it can significantly benefit from another round of revision. However, I won't object to accepting it if my co-reviewers champion it.

**Paper Topic And Main Contributions:**

The paper develops a benchmark called UP to support NLP for underrepresented languages. More specifically, it focuses on the evaluation of multilingual models in a few-shot setting (rather than a zero-shot setting) with respect to so-called user-centric tasks. These are tasks that users may encounter in their daily lives in the area of computer use. This concerns information access or the linking of different technologies via their data. The benchmark includes data for five tasks ("QA, OCR, autocomplete, semantic parsing and sentence-level transliteration"), two additional task setups (NER and retrieval), experimental setups focusing on ASR and MT, and baseline results contrasting subword and byte-based models. The focus of the benchmark is on scarce data rather than small data in a multilingual setting, focusing on under-represented languages and specific ("user-centric") tasks. In this way, it contrasts with related work. A key reference point for the development of the benchmark is the annotation cost. This relates to the amount of data that can be annotated in a given period of time (8 hours): it defines a lower boundary of the amount of annotation data to be considered for further processing. A second reference point is the efficiency in a multilingual computing scenario.

**Reasons To Accept:**

1. The paper provides a new benchmark with a special focus on underrepresented languages. This is very important because data for these languages is rare, while it is very important to develop NLP systems for all these languages. In this way, the benchmark can help to encourage such developments and to shift the focus from high-resource languages to low-resource languages.
2. The paper develops its benchmark for a wide range of different tasks, using a lot of data, augmenting it or generating new data (but with a focus on a certain anotation time limit). This broadens the applicability of the benchmark and makes it more likely to find followers who use this dataset for their NLP developments.
3. Successes in NLP and language modeling are probably due to the availability of sufficient data. It is interesting to see how this paper focuses on this task under its special time constraint (eight hours). More research is needed on what data is needed to at least provide a valuable resource that can be used for research on underrepresented languages, and how this is affected by annotation time limits. The paper provides some initial insights into this scenario.


**Reasons To Reject:**

1. The notion of a user-centric task (something like 'We focus on widely used, user-oriented tasks that benefit speakers of high-resource languages. resource languages') is not so clear. The authors presuppose a definition; a quantification would be more valid here.
2. The estimation of annotation effort, which is central to the paper, is underspecified: it is based on previous work and the authors' own 'annotation effort'. However, both of these reference points are not quantified, and especially not in a comparative way that would allow measurement in an inter-annotator estimation scenario. Here I would expect a more experimental setup to measure this important time, and I would also expect a deeper justification for the eight hours, which is certainly not that much, but why this time limit - just because it works sufficiently?
3. The UP benchmark is very rich in the coverage of tasks, but the description of the generation of datasets for each of these tasks is very small, so that a larger set of annotation efforts and data additions are noted, which would require more details to understand what was actually done. Each of these additions requires annotations (see, for example, lines 402-408), which in turn need to be evaluated in order to know something about the quality of the generated data and data additions. However, such evaluations (e.g. IAA coefficients) are not provided. Thus, the description of the tasks in Section 3 is manifold, but lacks a bit more information that would help to assess the success of the authors more reliably. This is all the more striking as the evaluation of the models in section 4 is not described as central (lines 452-454): apparently the evaluation of the data set extensions would have been more central. As it is, Section 3 is a very nice enumeration of selected tasks, but it is difficult for me to judge the exact results of data generation. The appendices could be more helpful in this respect too.
4. The paper focuses on few-shot learning and mentions the sensitivity of prompt design, but looks at single instructions per task and leaves the search for better designs to future work: this is a way around the underlying problem, but not a solution. Because of the centrality of few-shot learning to the paper, as opposed to zero-shot learning, one would expect a more sophisticated technique at this point.
5. The evaluation in section 4 is not very helpful in understanding the usefulness of the benchmark: is it just to say that there is now a new set of datasets with which we can show that some standard models perform poorly when trained with the "scarce" data from UP for some tasks - this is not so surprising since this data is by definition limited by the 8 hour annotation time (which is remarkably low). Or do we learn more about what it helps to use the UP data to improve systems that would otherwise not perform as well as they do when using this data from UP? Here the description is a bit short and does not fit perfectly into the overall picture of the paper. This has also to be criticized in the context of the lack of information on the annotation effort for each resource: if I don't know the quality of the 8-hour limited training splits, how do I know how fine-tuning based on this data affects the benchmarking of existing models? Again, the paper can be more informative in this respect. The discussion/analysis in Section 5 is also not so informative in this respect.

**Reproducibility:**

3: Could reproduce the results with some difficulty. The settings of parameters are underspecified or subjectively determined; the training/evaluation data are not widely available.

**Reviewer Confidence:**

3: Pretty sure, but there's a chance I missed something. Although I have a good feel for this area in general, I did not carefully check the paper's details, e.g., the math, experimental design, or novelty.

**Typos Grammar Style And Presentation Improvements:**

- "UP provides methodology" -> "UP provides a methodology"
- "QA Clark et al. (2020)" -> "QA (Clark et al. 2020)"
- "We focus on widely adopted user-facing tasks beneﬁting speakers of high-resource languages" -> "We focus on widely used, user-oriented tasks that benefit speakers of high-resource languages"
- "UP does not limit supervised training data in high-resource languages (HLs) while each under-represented language (UL) has a maximum of 8 hours of annotation effort in its training split;" -> "UP does not limit supervised training data in High Resource Languages (HLs), while a maximum of 8 hours of annotation effort is provided for each training split for each Underrepresented Language (UL);" - I am unsure about the quantification in the sentence; I make a proposal that is better understandable for me; here the authors should check their wording. It is also unclear that UP is limiting HLs while only  evaluating ULs.
- "In this way, we design for the task ﬁrst. " -> This sentence does not make sense for me. Do you mean "In this way, we first design the task."?
- "that aims ﬁll gaps" -> "that aims to ﬁll gaps"

---

> ### Author Rebuttal · Authors · 2023-08-28
>
> Thank you for the thoughtful feedback. We will aim to incorporate the suggestions and to clarify the under-specified settings.
>
> > 1. The notion of a user-centric task is not so clear
>
> Thank you for the suggestion. We would like to highlight that quantifying the overall usage or impact an NLP technology has is not straightforward. Jin et al. (2021; https://aclanthology.org/2021.findings-acl.273/) propose to measure the impact of applicable tools (e.g., MT) via the impact of Stage-4 technologies that build on them, such as Google Translate, Amazon Echo, etc., but do not provide any actual numbers. Blasi et al. (2022; https://aclanthology.org/2022.acl-long.376) similarly study “tasks that technology users interact with directly in their everyday life” including QA and MT but without any further quantification of the user base of each task.
> As information of all products that use a certain NLP technology is not available, we could use an estimate of the user base of the most widely used product where such information is available as a proxy. Note that also these numbers should be taken with a grain of salt. For MT, Google Translate has 1B users (https://blog.google/products/translate/one-billion-installs/). For QA and retrieval, Google Search processes 3.5B searches per day (https://www.internetlivestats.com/google-search-statistics/) with 1B+ users. We will attempt to provide a similar estimate of the usage of the other tasks in UP.
>
> > 2. The estimation of annotation effort, which is central to the paper, is underspecified
>
> We quantify and compare the annotation costs for each task in Table 2. We will provide further details regarding the specific measurements in a revised version of the paper. Regarding the selection of 8 hours as annotation time: We sought to select a realistic amount of annotation time that can be easily scaled up to extend to additional languages while still keeping the setting challenging for future research. In addition, some of the employed datasets do not provide much more than 8 hours of data per language, which limits the timeframes that can be used in a general setting.
>
> > 3. The UP benchmark is very rich in the coverage of tasks, but the description of the generation of datasets for each of these tasks is very small
>
> We had to limit the information on the creation of the new datasets due to space limitations in the submission. We will provide additional information on the data annotation of the new datasets in the revised version; many of these details are currently in the appendix and could be moved forward given additional space
>
> > 4. The paper focuses on few-shot learning and mentions the sensitivity of prompt design, but looks at single instructions per task and leaves the search for better designs to future work
>
> We use English as the prompt language, which has been shown to be the best source language for prompting current generative models in prior work (Winata et al., 2021, https://aclanthology.org/2021.mrl-1.1/; Lin et al., 2022, https://aclanthology.org/2022.emnlp-main.616/; Shi et al., 2023, https://arxiv.org/abs/2210.03057). These prior works similarly use a single instruction per task. This instruction may be machine-translated to other languages. Given the poor performance of MT for many of the languages in our study, we do not observe good performance with a machine-translated instruction. For this reason, we focus on the best-performing setting in accordance with prior work.
>
> > 5. The evaluation in section 4 is not very helpful in understanding the usefulness of the benchmark
>
> The fact that our models are fine-tuned in a joint multilingual setting means that while the data for each language may be low given the 8-hour annotation timeframe, we would expect that models that have learned strong multilingual representations would be able to benefit from positive cross-lingual transfer. In addition, recent LLMs have demonstrated exceptional performance even in 1-shot or 0-shot settings on many tasks; in contrast to this, we would expect much better performance even with 8 hours of annotated data.
>
> Our results clearly demonstrate that byte-based modeling is particularly important for under-represented languages with rich morphology while fine-tuning even on limited data outperforms in-context learning. We will provide further analysis regarding the performance of models across different resource groups in order to show a more differentiated picture of model performance.

---

### Official Review · Reviewer_jMj2 · 2023-08-05

**Soundness:** 3

**Excitement:**

3: Ambivalent: It has merits (e.g., it reports state-of-the-art results, the idea is nice), but there are key weaknesses (e.g., it describes incremental work), and it can significantly benefit from another round of revision. However, I won't object to accepting it if my co-reviewers champion it.

**Paper Topic And Main Contributions:**

This paper presents the UP benchmark, which is a user-centric scarce-data benchmark for under-represented languages. The main problem addressed by this paper is the lack of resources and evaluation benchmarks for under-represented languages, which makes it difficult to develop and evaluate natural language processing (NLP) models for these languages. The UP benchmark aims to fill this gap by providing a standardized evaluation framework for NLP models across nine tasks.

The main contributions of this paper are:

1. The development of a multilingual few-example benchmark.
2. Newly created data for five tasks.
3. Baseline results.

Overall, this paper makes contributions towards addressing the lack of resources and evaluation benchmarks for under-represented languages, and provides resources for researchers and practitioners working in this area.

**Questions For The Authors:**

1.What is the point of adding tasks and datasets directly from other sources directly to Up? I am referring to the tasks like ASR and MT.

2.How did you process the original MasakhaNER datasets? In appendix, you mentioned that "we process the data in order to align the token-level annotations with byte-level spans in the original pre-tokenized text." How did you do that?

**Reasons To Accept:**

The strengths of this paper, includes:

1. Large-scale evaluation: The UP benchmark evaluates the capabilites of language models accross 88 under-represented languages over 9 tasks.

2. New datasets: Creation of new datasets for OCR, auto-complete, question answering, semantic parsing, and transliteration

**Reasons To Reject:**

The limitations includes,

1.Including some datasets directly from other sources unnecessariy like Automatic speech recognition, Machine Translation. I do not get the merit of adding others datasets in UP without any contribution.

2.Following Joshi et al. (2020) they select languages in categories 1–3 (Line#132-135). However, there is no mention of these categories later. Catergory#1 is much low-resourced compared to category#3. Statistics of the datasets based on the categories, coverage of the tasks from the categories, presenting results by category-wise instead of just averaging would be more insightful, which are missing.


**Reproducibility:**

3: Could reproduce the results with some difficulty. The settings of parameters are underspecified or subjectively determined; the training/evaluation data are not widely available.

**Reviewer Confidence:**

3: Pretty sure, but there's a chance I missed something. Although I have a good feel for this area in general, I did not carefully check the paper's details, e.g., the math, experimental design, or novelty.

---

> ### Author Rebuttal · Authors · 2023-08-28
>
> Thank you for your feedback and for taking the time to review this work
>
> > 1.Including some datasets directly from other sources unnecessariy like ASR, MT
>
> We note that UP is primarily a benchmark (like SuperGLUE, XTREME, etc.); to accomplish its purpose of evaluating coverage of well-established language technologies in under-represented languages, we view the inclusion of these tasks as necessary. We included the best available resources we could find where possible, creating new resources where needed. That isBoth ASR and MT are such key technologies that have a wide range of use cases and make information much more accessible, which is crucial for under-represented languages. For both tasks, existing datasets already cover a large number of under-represented languages so we chose to prioritize tasks with sparser language coverage in our data collection efforts.
>
> > Statistics of the datasets based on the categories, coverage of the tasks from the categories, presenting results by categories is missing
>
> You can already view the language coverage of each task with each language’s resource level in Table 5 in the Appendix. We will add summary statistics per resource level to make inspecting this table easier. We will also present results per resource level to provide a more fine-grained breakdown.
>
> > 1.What is the point of adding tasks and datasets directly from other sources directly to UP?
>
> As mentioned above, the inclusion of MT and ASR is important for a user-centric evaluation and both tasks already have datasets with a large language coverage. We would also like to highlight that using existing datasets has been a common practice for prior benchmarks (see GLUE, SuperGLUE, XTREME, BUFFET, SUPERB, etc). We go significantly beyond prior benchmarks by creating new datasets for a wide range of applications.
>
> > 2.How did you process the original MasakhaNER datasets?
>
> We reproduce the preprocessing of the MasakhaNER datasets, which we re-apply to the original unlabeled source data. We then match each sentence in the unlabeled source data with its annotated and tokenized counterpart based on regular expressions and string matching. After finding a match, we map the annotated token-level spans to their corresponding spans in the original source text via regex matching. This automatic process finds matches for around 98% of cases. We resolve instances where no match is found through manual inspection and by consulting with the original dataset creators.

---

### Meta-Review · Area_Chair_mQGy · 2023-09-16

**Recommendation:** 4

**Metareview:**

All reviewers agree that this large-scale evaluation benchmark, which includes data and tasks for many under-resourced languages, will be useful for the multilingual NLP community as the current data and task coverage for these languages are scarce. In the rebuttal, it is made clearer what the notion of user-centric means (i.e., downstream tasks with a large number of end users), which justifies the selection of tasks including ASR and MT in the benchmark. The experimental setup, including the transformation of tasks into a seq-to-seq format and the few-shot prompting method, is also based on the recent standard practice of prompting language models. While one point of contention is about the choice of baselines, given the task format and the inclusion of multilingual models of different scales and different tokenization scheme, I believe the setup is sufficient. The authors also make salient their exact contributions to existing datasets by providing the table breakdown of their data contributions during the post-rebuttal discussion period, which hopefully will be included in the next revised version of the paper. One common weakness of the current paper raised by reviewers is the lack of in-depth analysis in Section 5; for instance, how the quality of the training splits of collected data (in a constrained setting of 8 hours) impacts the benchmarking evaluation and what is the reason for the poor performance of African languages.

---

### Decision · Program_Chairs · 2023-10-07

**Decision:**

Accept-Findings

**Comment:**

All reviewers agree that this large-scale evaluation benchmark, which includes data and tasks for many under-resourced languages, will be useful for the multilingual NLP community as the current data and task coverage for these languages are scarce. In the rebuttal, it is made clearer what the notion of user-centric means (i.e., downstream tasks with a large number of end users), which justifies the selection of tasks including ASR and MT in the benchmark. The experimental setup, including the transformation of tasks into a seq-to-seq format and the few-shot prompting method, is also based on the recent standard practice of prompting language models. While one point of contention is about the choice of baselines, given the task format and the inclusion of multilingual models of different scales and different tokenization scheme, I believe the setup is sufficient. The authors also make salient their exact contributions to existing datasets by providing the table breakdown of their data contributions during the post-rebuttal discussion period, which hopefully will be included in the next revised version of the paper. One common weakness of the current paper raised by reviewers is the lack of in-depth analysis in Section 5; for instance, how the quality of the training splits of collected data (in a constrained setting of 8 hours) impacts the benchmarking evaluation and what is the reason for the poor performance of African languages.